# Phorbolester-activated Munc13-1 and ubMunc13-2 exert opposing effects on dense-core vesicle secretion

Sébastien Houy[1†], Joana S Martins[1†], Noa Lipstein[2,3], Jakob Balslev Sørensen[1*]

[1]Department of Neuroscience, University of Copenhagen, Copenhagen, Denmark; [2]Department of Molecular Neurobiology, Max-Planck-Institute for Multidisciplinary Sciences, Göttingen, Germany; [3]Leibniz-Forschungsinstitut für Molekulare Pharmakologie (FMP), Berlin, Germany

**Abstract** Munc13 proteins are priming factors for SNARE-dependent exocytosis, which are activated by diacylglycerol (DAG)-binding to their C1-domain. Several Munc13 paralogs exist, but their differential roles are not well understood. We studied the interdependence of phorbolesters (DAG mimics) with Munc13-1 and ubMunc13-2 in mouse adrenal chromaffin cells. Although expression of either Munc13-1 or ubMunc13-2 stimulated secretion, phorbolester was only stimulatory for secretion when ubMunc13-2 expression dominated, but inhibitory when Munc13-1 dominated. Accordingly, phorbolester stimulated secretion in wildtype cells, or cells overexpressing ubMunc13-2, but inhibited secretion in Munc13-2/*Unc13b* knockout (KO) cells or in cells overexpressing Munc13-1. Phorbolester was more stimulatory in the Munc13-1/*Unc13a* KO than in WT littermates, showing that endogenous Munc13-1 limits the effects of phorbolester. Imaging showed that ubMunc13-2 traffics to the plasma membrane with a time-course matching $Ca^{2+}$-dependent secretion, and trafficking is independent of Synaptotagmin-7 (Syt7). However, in the absence of Syt7, phorbolester became inhibitory for both Munc13-1 and ubMunc13-2-driven secretion, indicating that stimulatory phorbolester x Munc13-2 interaction depends on functional pairing with Syt7. Overall, DAG/phorbolester, ubMunc13-2 and Syt7 form a stimulatory triad for dense-core vesicle priming.

**\*For correspondence:**
jakobbs@sund.ku.dk

[†]These authors contributed equally to this work

**Competing interest:** The authors declare that no competing interests exist.

## Editor's evaluation

This fundamental study study reveals that phorbol esters have a stimulatory effect on chromaffin cell secretion via ubMunc13-2 but an inhibitory effect via Munc13-1. These convincingly demonstrated, opposing effects of the two closely related Munc13 paralogs are surprising and, although it remains unclear how these findings relate to the mechanism of synaptic vesicle release, the study reveals important differences between the two isoforms of this central priming protein.

## Introduction

Release of chemical neurotransmitters by exocytosis of small synaptic vesicles (SV) forms the basis for rapid communication between neurons, whereas larger dense core vesicles (DV) contain neuropeptides necessary for neuromodulation. The machinery for exocytosis of SVs and DVs must incorporate both a basic lipid fusion apparatus, and layers of control that enable vesicular release to be restricted in time and space. The SNARE-complex, formed between vesicular VAMP/synaptobrevin, and plasma membrane located syntaxin-1 and SNAP-25, is the canonical fusion machinery (*Rizo, 2018*), and its mutation leads to complex neurodevelopmental disorders (*Verhage and Sørensen, 2020*). SNARE-complex formation is widely believed to coincide with vesicle priming, the process that renders the

vesicle releasable (*Sørensen et al., 2006*). Upstream processes regulating SNARE-complex formation therefore become priming factors *par excellence*; this includes Munc18-1 and Munc13 proteins. Munc18-1 binds to an inactivated, 'closed' syntaxin-1 configuration (*Dulubova et al., 1999*). Munc13s facilitate the opening of syntaxin-1 through their catalytic MUN-domain (*Ma et al., 2011*; *Richmond et al., 2001*; *Yang et al., 2015*), which allows Munc18-1 to act as a template for SNARE-complex formation (*Baker et al., 2015*; *Kalyana Sundaram et al., 2021*; *Parisotto et al., 2014*; *Shu et al., 2020*; *Sitarska et al., 2017*). Synaptotagmin-1 binds to $Ca^{2+}$ and triggers release in an interplay with complexin and the SNAREs themselves (*Mohrmann et al., 2015*; *Südhof, 2013*). Synaptotagmin-7 (Syt7) is a slower $Ca^{2+}$-sensor, which is able to both trigger slow release downstream of SNARE-complex formation (*Bacaj et al., 2013*; *Schonn et al., 2008*), but also to act upstream of complex formation in a role leading to $Ca^{2+}$-dependent vesicle priming (*Liu et al., 2014*; *Tawfik et al., 2021*). The exocytotic cascade culminates in the formation of a fusion pore through which water-soluble signaling molecules can escape (*Álvarez de Toledo et al., 2018*; *Chang et al., 2017*; *Sharma and Lindau, 2018*).

The vesicle priming machinery incorporates several points of regulation, triggered by receptor activation, or intracellular $Ca^{2+}$. One of them is the binding of DAG to the C1-domain in Munc13 proteins (*Betz et al., 1998*), often studied using β-phorbolesters, which are DAG-mimics that potentiate release from chromaffin cells (*Smith et al., 1998*) and central synapses (*Lou et al., 2005*; *Malenka et al., 1986*; *Shapira et al., 1987*). Phorbolester targets Munc13s to the plasma membrane (*Ashery et al., 2000*; *Betz et al., 1998*). Upon mutation to unfold the C1-domain of Munc13-1 (H567K), phorbolester does not potentiate glutamatergic neurotransmission in cultured hippocampal neurons (*Rhee et al., 2002*). Later experiments showed that both the C1-domain and the $Ca^{2+}$-unbound C2B-domain are inhibitory (*Li et al., 2019*; *Michelassi et al., 2017*), but inhibition can be relieved by $Ca^{2+}$-binding to the C2B-domain, or binding of DAG to C1 (*Michelassi et al., 2017*). Structural studies have identified a ~20 nm elongated rod-like structure formed by the C1-C2B-MUN domain (*Xu et al., 2017*). The C-terminal C2C-domain on the other side of this rod binds to membranes, but not to $Ca^{2+}$. This makes it possible for the C1-C2B-MUN-C2C structure to bridge the plasma and vesicular membranes, which is necessary for neurotransmitter release (*Liu et al., 2016*; *Padmanarayana et al., 2021*; *Quade et al., 2019*). Bridging can take place at different angles, depending on the binding mode at the plasma membrane; one binding mode uses a polybasic face and results in an upright/perpendicular orientation that hinders release, whereas another binding mode involves DAG-$Ca^{2+}$-$PIP_2$-binding by the C1-C2B-domain, which results in a slanted orientation that facilitates release (*Camacho et al., 2021*). $Ca^{2+}$/phospholipid-binding to the C2B-domain accelerates SV recruitment and reduces synaptic depression (*Lipstein et al., 2021*). Similar effects are seen upon calmodulin (CaM)-binding, with different penetrance in different synapses (*Junge et al., 2004*; *Lipstein et al., 2013*).

Munc13 proteins exist as different paralogs with overlapping functions. Munc13-1 is critically involved in synaptic vesicle release, with glutamate release arrested in most synapses in the absence of Munc13-1 (*Augustin et al., 1999b*). Munc13-2 exists as two different isoforms, due to alternative promotors; ubMunc13-2 is ubiquitously expressed, whereas bMunc13-2 is brain specific (*Brose et al., 1995*; *Kawabe et al., 2017*). Recently, mutations in Munc13-2 were linked to human epilepsy (*Wang et al., 2021a*). bMunc13-2 is expressed in a subset of glutamatergic synapses (*Kawabe et al., 2017*), and shapes paired pulse ratio and frequency facilitation at the hippocampal mossy fiber synapse (*Breustedt et al., 2010*). In synapses formed by pyramidal cells on GABAergic interneurons in the CA1 region of the hippocampus both Munc13-1 and Munc13-2 are expressed, but no consequences were identified upon Munc13-2 deletion (*Holderith et al., 2021*). Similarly, Munc13-2 deletion was without consequence in the Calyx of Held synapse (*Chen et al., 2013*) or the mouse photoreceptor ribbon synapse (*Cooper et al., 2012*). Munc13-1 is also involved in insulin secretion (*Kang et al., 2006*; *Kwan et al., 2006*; *Sheu et al., 2003*). In neurons, elimination of both Munc13-2 and Munc13-1 reduced the release of DVs by approximately 60%, whereas overexpression of Munc13-1 specifically increased DV secretion at extrasynaptic sites (*van de Bospoort et al., 2012*). ubMunc13-2 is involved in release of adrenaline/noradrenaline from adrenal chromaffin cells, as its hyper- and hypo-expression causes correlated changes in release (*Man et al., 2015*). Adrenal chromaffin cells also express Munc13-1, but no consequences were identified upon its deletion (*Man et al., 2015*). In an in vitro fusion assay, the C1-C2-MUN domain of Munc13-1 strongly stimulated fusion of synaptic vesicles, sped up fusion of

insulin granules, but had no effect on DVs from PC12-cells (*Kreutzberger et al., 2017*; *Kreutzberger et al., 2019*).

Here, we set out to understand the different roles of Munc13-1 and ubMunc13-2 co-expressed in adrenal chromaffin cells (*Man et al., 2015*). Munc13-1 and ubMunc13-2 both contain DAG/phorbolester-binding C1-domains. We surprisingly find that phorbolester can be positive or negative for DV secretion depending on the expressed Munc13 paralog. The stimulatory effect of phorbolester on DV fusion depends on the co-expression of ubMunc13-2 and Syt7 in the same cell, identifying a stimulatory triad of ubMunc13-2, Syt7 and DAG/phorbolester for DV priming.

## Results

Mouse adrenal chromaffin cells have proven useful for deciphering the molecular basis of neurotransmitter release (*Neher, 2018*). Being small and compact, they are ideal for patch-clamp capacitance measurements, and release can be stimulated rapidly by $Ca^{2+}$-uncaging (*Houy et al., 2021*), which bypasses $Ca^{2+}$-influx, allowing a focus on the release machinery. $Ca^{2+}$-uncaging empties the primed vesicle pools and allows simultaneous determination of vesicle pool sizes and fusion rates. Mouse adrenal chromaffin cells express both Munc13-1 and ubMunc13-2, encoded by the genes *Unc13a* and *Unc13b*, respectively. Previous work showed that Munc13-1 overexpression in bovine chromaffin cells (*Ashery et al., 2000*; *Betz et al., 2001*), in mouse WT cells (*Stevens et al., 2005*) or *Unc13a/Unc13b* double knockout (KO) chromaffin cells (*Man et al., 2015*) increased secretion, but ubMunc13-2 appeared even more potent (*Man et al., 2015*). Genetic deletion of Munc13-2 in mouse chromaffin cells markedly reduced secretion, whereas no effect was detected after deletion of Munc13-1 (*Man et al., 2015*). These findings support the notion that Munc13-1 and ubMunc13-2 play overlapping roles, with ubMunc13-2 dominating in wildtype cells, but the effect of phorbolester was not investigated.

### Munc13-2 is required for the stimulatory effect of phorbolesters in chromaffin cells

To understand the molecular requirements for the well-known ability of phorbolester to stimulate secretion in chromaffin cells (*Nagy et al., 2006*; *Smith et al., 1998*), we applied Phorbol 12-Myristate 13-Acetate (PMA, 0.1 µM, applied for 5–60 min) to chromaffin cells isolated from newborn mouse *Unc13b* KO and WT littermates (*Varoqueaux et al., 2002*). We stimulated secretion by calcium uncaging, and monitored exocytosis by parallel capacitance and amperometric measurements. As expected, application of PMA doubled the secretion amplitude in *Unc13b* WT cells, whether measured by capacitance or amperometry (*Figure 1A*; traces are the mean of all measured cells). The pre-stimulation calcium concentration was ~0.9 µM, which is near the optimal concentration for calcium-dependent vesicle priming in these cells (*Pinheiro et al., 2013*; *Tawfik et al., 2021*). We performed kinetic analysis of the capacitance traces by fitting them with a sum of two exponential functions and a straight line, using an automatic fitting routine (*Tawfik et al., 2021*). This resulted in the determination of the two primed vesicle pools, which are denoted the Readily Releasable Pool (RRP), and the Slowly Releasable Pool (SRP). Both pools were increased significantly in size by PMA treatment (*Unc13b* WT, RRP = 88.1 fF±19.5 fF (mean ± SEM),+PMA, RRP = 232.2 fF±35.9 fF, Mann-Whitney test: *P*=0.0001, *Unc13b* WT, SRP = 48.1 fF±7.8 fF,+PMA, SRP = 110.4 fF±15.9 fF, Mann-Whitney test: p=0.0009; *Figure 1B and E*). The sustained phase of release was not significantly changed by PMA (*Unc13b* WT, sustained = 123.6 fF±19.8 fF,+PMA, sustained = 187.4 fF±31.3 fF, Mann-Whitney test: p=0.1641, *Figure 1D*). The kinetics of RRP release was slightly slower in the presence of PMA (*Unc13b* WT, $\tau_{fast}$ = 23.0 ms,+PMA, $\tau_{fast}$ = 30.3 ms, Mann-Whitney test: p=0.0265; *Figure 1C*), whereas the kinetics of SRP release was unchanged (*Figure 1F*).

Strikingly, when we applied PMA to cultures of chromaffin cells prepared from *Unc13b* KO littermates, the effect of PMA was strongly inhibitory (*Figure 1G*), leading to a factor ~3 reduction in overall release. Kinetic analysis revealed significant reductions in RRP, SRP and sustained components (*Unc13b* KO, RRP = 47.9 fF±14.4 fF,+PMA, RRP = 13.6 fF±5.1 fF, Mann-Whitney test: p=0.0405, *Unc13b* KO, SRP = 56.0 fF±10.5 fF,+PMA, SRP = 16.5 fF±3.7 fF, Mann-Whitney test: p=0.0016, *Unc13b* KO, sustained = 26.2 fF±7.0 fF,+PMA, sustained = 14.4 fF±4.4 fF, Mann-Whitney test: p = 0.0496 *Figure 1H, K and J*). The kinetics of RRP and SRP secretion remained unchanged (*Figure 1I*

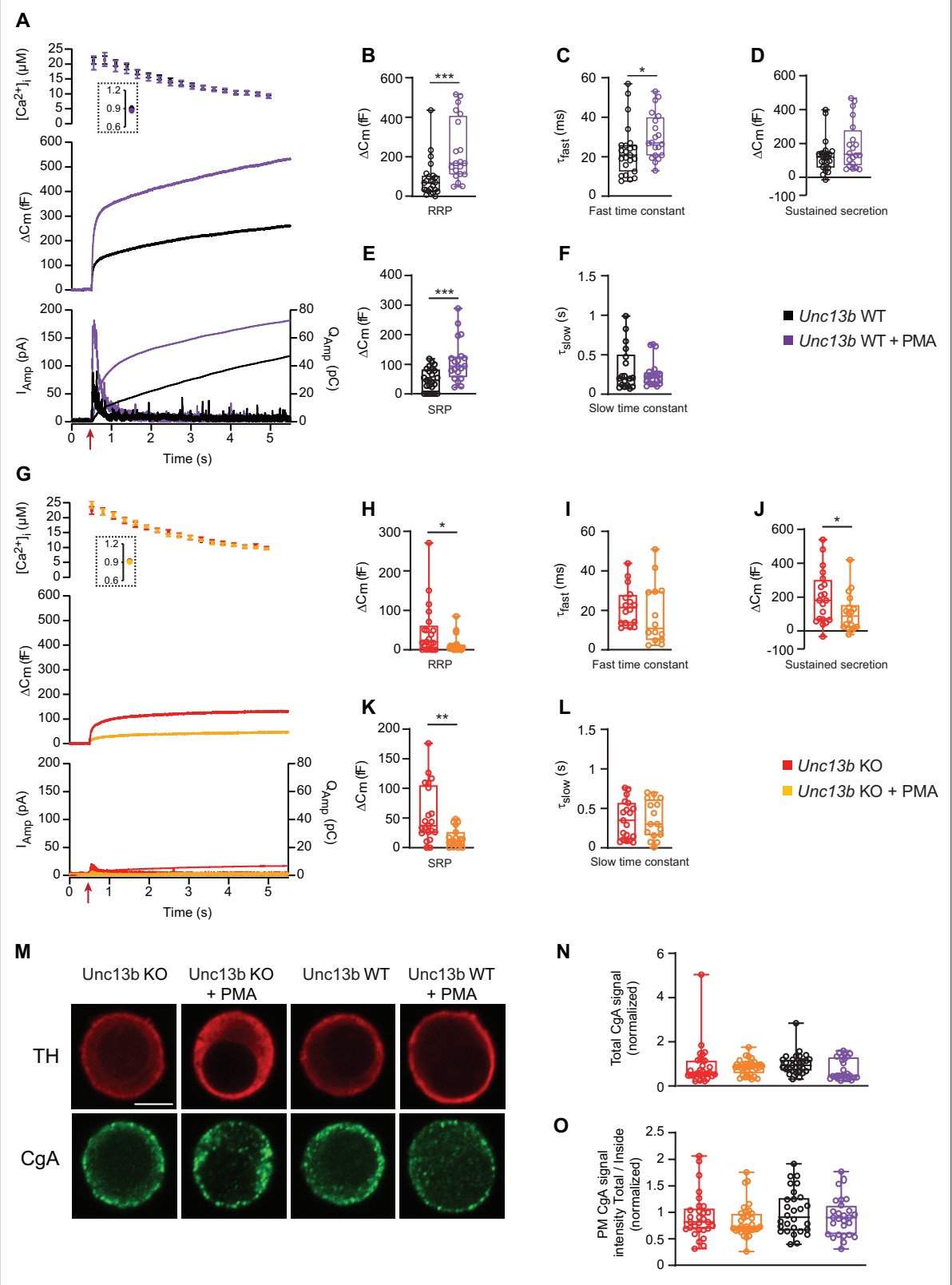

**Figure 1.** In the absence of Munc13-2, phorbolesters are inhibitory for dense-core vesicle secretion. (**A**) Calcium uncaging experiment in *Unc13b* WT chromaffin cells from newborn mice (P0–P2) in the absence and presence of PMA (Black and purple traces). Top panel: [Ca²⁺] before (insert) and after calcium uncaging (uncaging flash at red arrow, bottom panel). Middle panel: capacitance traces (mean of all cells) show that PMA treatment potentiates secretion in WT cells (higher amplitude). Bottom panel: Mean amperometry (left ordinate axis) and mean integrated amperometry (right ordinate

*Figure 1 continued on next page*

*Figure 1 continued*

axis). (**B**) Sizes of the RRP. (**C**) Time constants of fusion for fast (i.e. RRP) secretion. (**D**) Sustained secretion (secretion during the last 4 s of the trace in A). (**E**) Sizes of the SRP. (**F**) Time constants of fusion for slow (i.e. SRP) secretion. (**G**) Calcium uncaging experiment in *Unc13b* KO in the absence and presence of PMA (red and orange traces). Panels are arranged as in A. (**H**) Sizes of the RRP. (**I**) Time constants of fusion for fast secretion. (**J**) Sustained secretion (secretion during the last 4 s of the trace in G). (**K**) Sizes of the SRP. (**L**) Time constants of fusion for slow secretion. Data information: In panel A and G (top) data are presented as mean ± SEM; in A and G (middle and bottom panels), the traces are the mean of all cells. In B-F and H-L, data are presented as box and whiskers. *: p<0.05; **: p<0.01; ***: p<0.001. Mann-Whitney test comparing WT cells with WT cells treated with PMA, or KO cells with KO cells treated with PMA or. Number of cells, KO: n=21 cells; KO +PMA: n=19 cells; WT: n=24 cells; WT +PMA: n=20 cells. (**M**) Single confocal slices of *Unc13b* KO and WT mouse chromaffin cells with or without PMA stained against tyrosine hydroxylase (α-TH) and Chromogranin A (α-CgA). Scale bar: 5 µm. (**N**) normalized total cellular CgA fluorescence (integrated density) in Unc13b KO and WT cells with or without PMA. Kruskal-Wallis test with Dunn's post-hoc test: no differences were significant. (**O**) Quantification of the plasma membrane fluorescence intensity of CgA (total divided by inside fluorescence, using integrated density) in Unc13b KO and WT cells with or without PMA (normalization to Unc13b WT). Data information: quantitative data are presented as box and whiskers. In (**M–O**) number of cells, KO: n=30 cells; KO +PMA: n=32 cells; WT: n=29 cells; WT +PMA: n=28 cells.

The online version of this article includes the following source data for figure 1:

**Source data 1.** Quantitative data.

___

*and L*). Importantly, to protect against variance between cultures, measurements with and without PMA were obtained from the same cultures originating from single animals and measured on the same days. Opposite effects of PMA on *Unc13b* WT and KO were detected using the same batches of PMA.

Lack of secretion upon Ca$^{2+}$-uncaging could be expected if PMA-treatment of *Unc13b* KO cells led to spontaneous release, depleting the cells of secretory vesicles. To rule out this possibility, we stained *Unc13b* WT and KO adrenal chromaffin cells against Tyrosine Hydroxylase (TH, a marker of adrenal chromaffin cells) and chromogranin A, a releasable component of DVs. All four groups (*Unc13b* WT, *Unc13b* WT +PMA, *Unc13b* KO, *Unc13b* KO +PMA) displayed strong chromogranin staining (*Figure 1M–N*), with a tendency towards lower staining in the WT +PMA group (p=0.1084), that is the group with the highest secretion (*Figure 1A*). Quantifying the amount of CgA staining close to the membrane also did not result in significant differences (*Figure 1O*). Thus, Unc13b KO cells have intact secretory potential with and without PMA.

These data show that in mouse adrenal chromaffin cells, ubMunc13-2 is required for phorbolester to stimulate secretion, and in its absence, phorbolester is inhibitory. Phorbolesters also activate Protein kinase C (PKC), which stimulates secretion, but apparently this effect does not prevail in the absence of ubMunc13-2, probably due to an overall weak PKC-dependence of secretion from newborn or embryonic mouse chromaffin cells (see Discussion).

## Phorbolesters are more potent in the absence of Munc13-1

We next investigated the consequence of deleting Munc13-1 for the effect of PMA. To this end, we performed measurements in *Unc13a* WT and KO littermate cultures. Strikingly, PMA stimulation of *Unc13a* KO cells led to a higher overall secretion level than in *Unc13a* WT cells treated with PMA (*Figure 2A*, traces are mean of all measured cells). Kinetic analysis showed that PMA stimulated the RRP size in both *Unc13a* WT and Unc13a KO (*Unc13a* WT, RRP = 36.9 fF±7.0 fF,+PMA, RRP = 88.5 fF±11.9 fF, *Unc13a* KO, RRP = 44.4 fF±7.7 fF,+PMA, RRP = 169.2 fF±37.9 fF, Kruskal-Wallis test – Dunn's multiple comparisons test WT vs WT PMA p=0.0091; KO vs KO PMA p=0.0288, *Figure 2B*). On average, the RRP was larger in the *Unc13a* KO +PMA than in the *Unc13a* WT +PMA (169.2 fF ±37.9 fF vs RRP = 88.5 fF±11.9 fF), but the difference did not reach statistical significance. The SRP-size was significantly larger in the Unc13a KO +PMA than in the Unc13a WT +PMA group (*Unc13a* WT, SRP = 29.8 fF±7.9 fF,+PMA, SRP = 45.4 fF±7.0 fF, *Unc13a* KO, SRP = 40.5 fF±9.5 fF,+PMA, SRP = 95.6 fF±14.2 fF, Kruskal-Wallis test – Dunn's multiple comparisons test: KO vs KO PMA p=0.0048; WT PMA vs KO PMA p=0.0437; WT vs KO PMA p=0.0003, *Figure 2E*). The sustained release and the kinetics of RRP and SRP fusion were not significantly different between conditions (*Figure 2C, D and F*). The hypothesis that PMA has a larger effect in Unc13a KO than in Unc13a WT can also be tested by arranging the data in a two-way ANOVA, with genotype and drug application (with or without PMA) as orthogonal factors. The hypothesis then becomes identifiable as a significant interaction between the two factors. Indeed, performing this analysis showed that the interaction was close to significant in

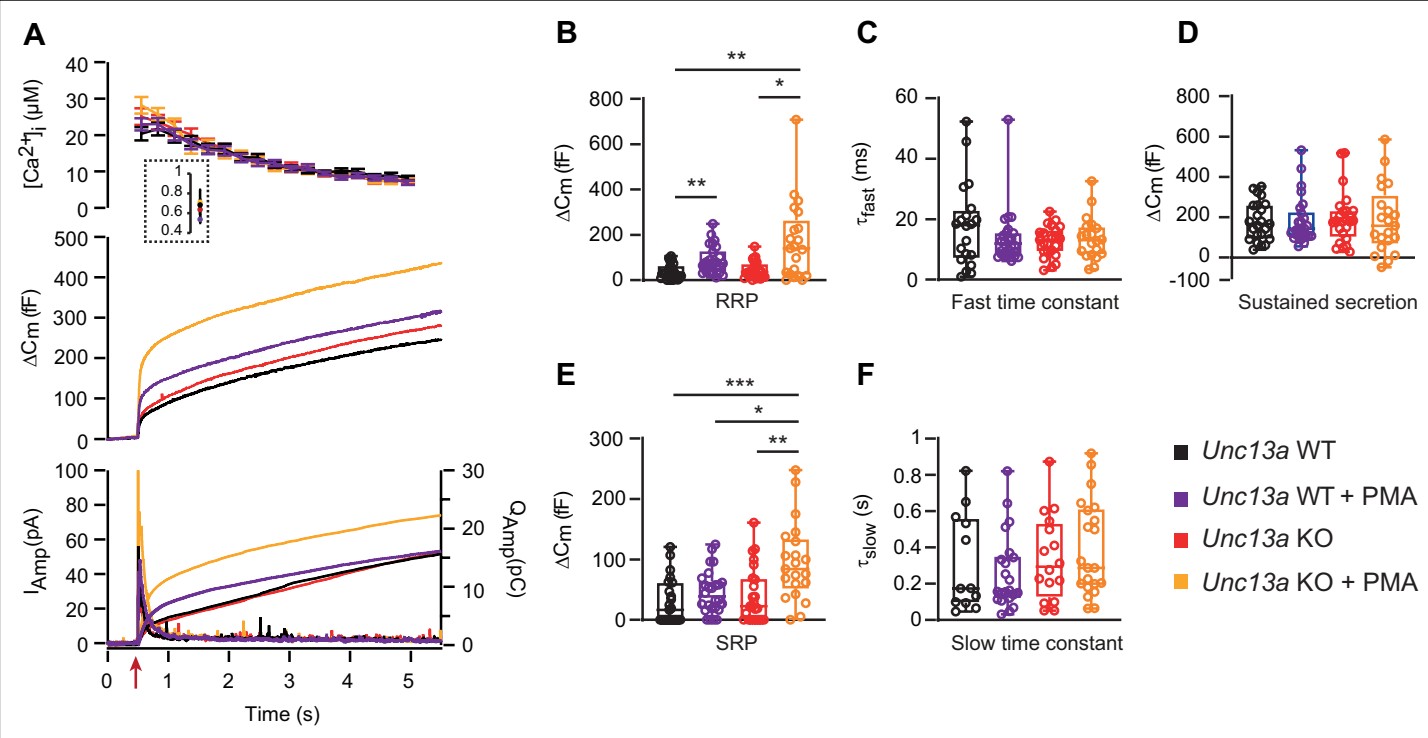

**Figure 2.** Absence of Munc13-1 potentiates phorbolester-induced secretion. (**A**) Calcium uncaging experiment in *Unc13a* WT (black traces) and KO embryonic (E18) chromaffin cells (red traces) untreated or treated with PMA (WT-PMA: Blue traces; KO-PMA: Yellow traces). Panels are arranged as in *Figure 1A*. (**B**) Sizes of the RRP. (**C**) Time constants of fusion for fast (i.e. RRP) secretion. (**D**) Sustained secretion. (**E**) Sizes of the SRP. (**F**) Time constants of fusion for slow (i.e. SRP) secretion. In the absence of Munc13-1 (*Unc13a* KO), PMA potentiates secretion more than in *Unc13a* WT cells. Data information: In panel A (top panel), data are presented as mean ± SEM; in A (middle and bottom panels), the traces are the mean of all cells. In B-F, data are presented as box and whiskers. * p<0.05; ** p<0.01. Kruskal-Wallis test with Dunn's post-hoc test. Number of cells in (**A–F**): WT: n=23 cells; KO: n=24 cells; WT +PMA: n=26 cells; KO +PMA: n=21 cells.

The online version of this article includes the following source data for figure 2:

**Source data 1.** Quantitative data.

both cases (p=0.0562 for RRP size and p=0.0442 for SRP size), indicating a likely interaction between the two factors, which was again significant for the SRP.

When comparing the two mouse lines, the secretory amplitude was larger in the *Unc13b* WT than in *Unc13a* WT, especially in the presence of PMA (*Figure 2A* vs *Figure 1A*). However, *Unc13a* WT and *Unc13b* WT cultures were not prepared in parallel. Moreover, because *Unc13a* is perinatally lethal (*Augustin et al., 1999a*) we used embryonic animals (18. embryonic day) for *Unc13a* KO and WT, whereas for *Unc13b* WT and KO, we used postnatal (P0-P2) animals (Materials and Methods). The data sets are therefore not directly comparable.

Overall, phorbolester is more potent in the absence of Munc13-1, consistent with the notion that functional interaction of phorbolester with endogenous Munc13-1 interaction is negative for secretion in chromaffin cells.

## Overexpressed Munc13-1 and Munc13-2 display different trafficking and phorbolester effects

Next, we overexpressed each Munc13 paralog (Munc13-1, ubMunc13-2) separately, and investigated whether the effect of phorbolester would differ when either paralog would dominate the cell. These experiments made use of the Munc13-1-EGFP and ubMunc13-2-EGFP Semliki Forest Virus (SFV) constructs previously used for expression in bovine or mouse chromaffin cells (*Ashery et al., 2000*; *Man et al., 2015*; *Zikich et al., 2008*). We took advantage of the EGFP-tag to visualize trafficking of Munc13 following Ca²⁺-uncaging using a CCD camera (*Figure 3—figure supplement 1*). Thus, before and after the uncaging flash, cells were illuminated by 488 nm light to visualize EGFP. This

was possible using the same (fura-containing) pipette solution, because 488 nm light does not excite fura dyes. Although in these measurements we did not measure $[Ca^{2+}]_i$, we could measure $[Ca^{2+}]$ in separate experiments using identical flash lamp setting, dichroic mirror, objective, and pipette solution (see Materials and Methods). Thereby, we established that these measurements led to a pre-stimulation $[Ca^{2+}]_i$ = 0.64 µM±0.09 µM and a post-stimulation $[Ca^{2+}]_i$ = 26.8 µM±1.8 µM (means ± SEM, n=14) (*Figure 3—figure supplement 2*). Some measurements might deviate from this range. Successful expression of Munc13-1-EGFP and ubMunc13-2-EGFP was verified by Western blotting of SFV-infected HEK-cells (*Figure 3—figure supplement 3A*), and quantification of EGFP fluorescence from single chromaffin cells (*Figure 3—figure supplement 3B*). We imaged Munc13-1-EGFP, ubMunc13-2-EGFP and the H567K-mutation of Munc13-1-EGFP, which does not bind to phorbolesters (*Betz et al., 1998*), while applying PMA (*Figure 3—figure supplement 4*). As expected, Munc13-1-EGFP and ubMunc13-2-EGFP, but not Munc13-1-EGFP H567K translocated to the plasma membrane over the course of several minutes. This time course most likely reflects the time it takes for PMA to penetrate the cell, combined with the time it takes for Munc13 to diffuse to the plasma membrane in the absence of increased $[Ca^{2+}]$. Since the capacitance measurements above indicated that the main effect of PMA is on secretion amplitude, not kinetics (see also *Nagy et al., 2006*), we simplified the analysis and only distinguished between burst secretion (first 1 s secretion after $Ca^{2+}$ uncaging, corresponding approximately to RRP and SRP fusion) and sustained secretion (last 4 s of secretion), as well as total secretion (the sum of burst and sustained release).

Expression of ubMunc13-2-EGFP in *Unc13b* KO cells resulted in large secretory amplitude, ~800 fF (*Figure 3B*). In response to PMA, secretion was increased even further, resulting in an increase in burst size, total and sustained secretion; the two former effects were statistically significant (Burst - Overexpressed (OE) ubMunc13-2-EGFP: 239.09 fF ±25.96 fF,+PMA: 628.72 fF ±119.64 fF, Mann-Whitney test: p=0.0314; Total - OE ubMunc13-2 EGFP: 808.93 fF ±107.82 fF,+PMA: 1344.53 fF ±178.70 fF, Mann-Whitney test: p=0.0479; Sustained - OE ubMunc13-2-EGFP: 570.22 fF ±87.67 fF; 714.66±89.75, Mann-Whitney test: p=0.2992; *Figure 3D*). ubMunc13-2-EGFP was present in the cytosol (*Figure 3A* top panels), but upon uncaging, part of the protein trafficked to the plasma membrane within a few seconds, with a kinetics roughly matching sustained release - in *Figure 3B* the normalized PM fluorescence was replotted behind the capacitance trace to allow a comparison (*Figure 3A–B*; *Figure 3—figure supplement 5A*). Upon application of PMA, ubMunc13-2-EGFP was already at the plasma membrane and did not traffic further upon $Ca^{2+}$ increase (*Figure 3A* bottom panels, *Figure 3B*, *Figure 3—figure supplement 5B*). To test the significance of the $Ca^{2+}$-dependent trafficking, we could use the trafficking index at 5 s compared to preflash values. However, in principle the uncaging flash itself could change our measure of PM localization (total fluorescence divided by fluorescence in the cytosol), because it represents a strong flash of ultraviolet light that could lead to bleaching of both EGFP and background fluorescence. If bleaching of EGFP and background are not exactly proportional, the measure of PM localization could change even in the absence of trafficking. Therefore, we compared trafficking at 5 s after uncaging in the absence of PMA to the PMA-group, since the PMA group did not show significant trafficking by virtue of being already at the plasma membrane. This showed that $Ca^{2+}$-dependent trafficking was highly significant (Mann-Whitney test: p<0.0001; *Figure 3C*).

Expression of Munc13-1-EGFP in *Unc13a* KO cells resulted in approximately 400 fF secretion (*Figure 4B*), more than in the *Unc13a* WT (*Figure 2*), although we here compare experiments not carried out in parallel. Munc13-1-EGFP localized to the cytosol, with part of the protein found in larger accumulations in some, but not all cells (compare different examples in *Figure 4A*, *Figure 5A*, *Figure 4—figure supplement 1A*, *Figure 7—figure supplement 1*). We speculated that the accumulations might be related to the higher average expression level of Munc13-1 compared to ubMunc13-2, as assessed by cellular EGFP fluorescence (*Figure 3—figure supplement 3B*). Expression of the Munc13-1-EGFP for shorter times (than the standard 12–17 hr, Materials and methods) was attempted, but this was not successful because of a lack of consistent expression as detected by EGFP fluorescence. Upon PMA application, part of the protein localized to the plasma membrane (*Figure 4A*; *Figure 3—figure supplement 4*). As a further indication that Munc13-1-EGFP is functional, we expressed it in cells from CD1-mice (an outbred mouse strain). Expression caused a clear and statistically significant increase in total and sustained secretion (*Figure 4—figure supplement 1*). We verified the sequence of both the Munc13-1-EGFP and

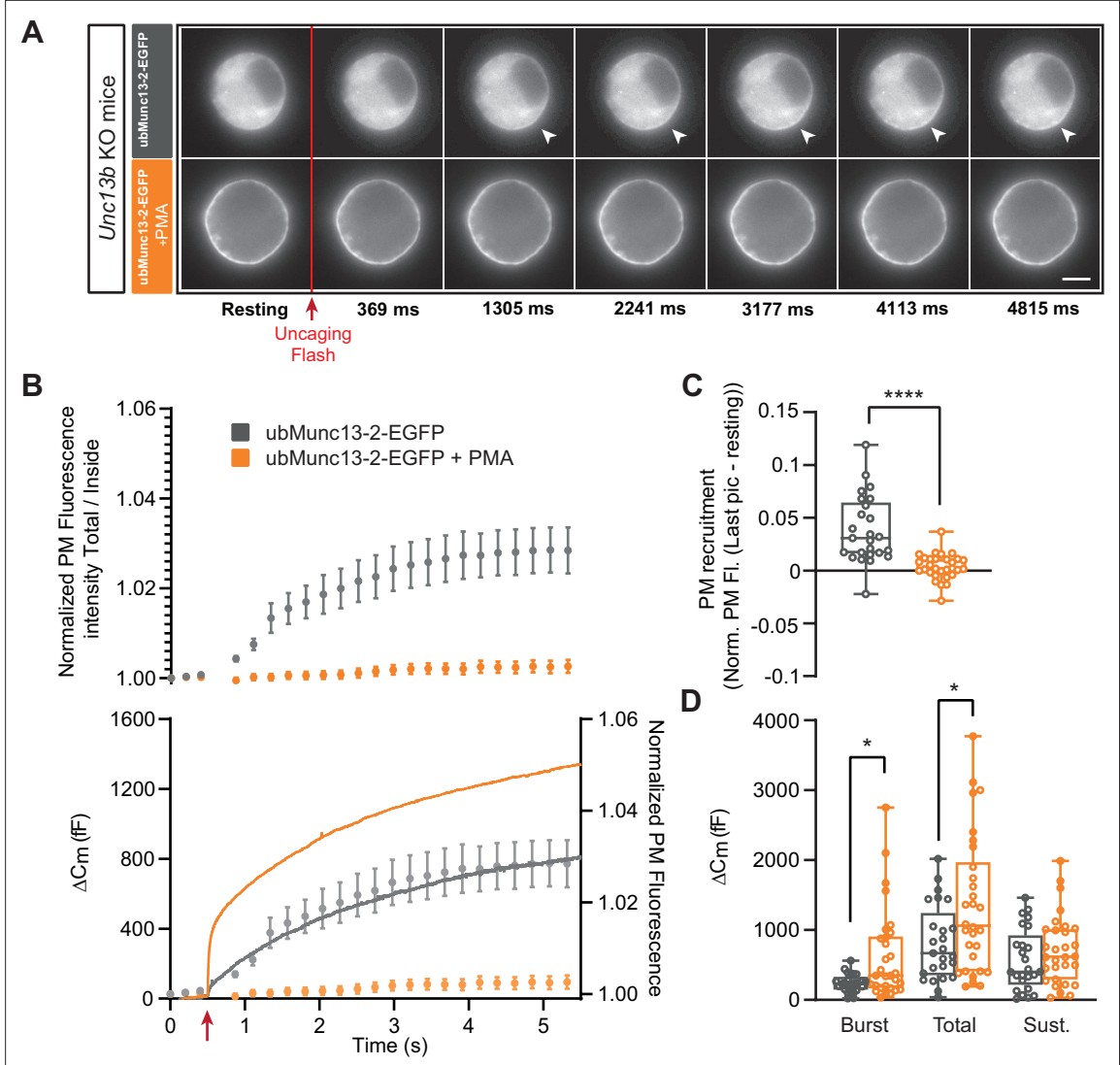

**Figure 3.** UbMunc13-2-dependent secretion is potentiated by phorbolester. (**A**) Wide-field imaging of ubMunc13-2-EGFP expressed in *Unc13b* KO chromaffin cells and treated with PMA (orange) or left untreated (grey). ubMunc13-2 was recruited at the plasma membrane after calcium uncaging (Top panel - white arrows) and upon PMA treatment (bottom panels). The uncaging light flash is represented by a red arrow. Scale bar: 5 μm. (**B**) Top: Quantification of the plasma membrane ubMunc13-2-EGFP fluorescence intensity normalized to the first picture acquired in resting condition (mean ± SEM of all cells). Bottom: Capacitance traces (mean of all cells) obtained simultaneously with EGFP imaging, showing that PMA treatment potentiates secretion in *Unc13b* KO cells expressing ubMunc13-2. Right axis: Normalized PM fluorescence replotted from panel (**A**) showing that the time course coincides with the capacitance trace. (**C**) Plasma membrane ubMunc13-2-EGFP recruitment (total fluorescence / inside fluorescence) normalized to resting (pre-stimulation) values. (**D**) Sizes of Burst, Total and Sustained release. Data information: data in panel B are presented as mean ± SEM; data in panels C and D are presented as box and whiskers. *: p<0.05; ****: p<0.0001, Mann-Whitney tests. ubMunc13-2-EGFP in *Unc13b* KO: n=25 cells; ubMunc13-2-EGFP in *Unc13b* KO +PMA: n=30 cells.

The online version of this article includes the following source data and figure supplement(s) for figure 3:

**Source data 1.** Quantitative data.

**Figure supplement 1.** Experimental setup combining capacitance measurements and GFP-imaging in adrenal chromaffin cells.

**Figure supplement 2.** Calcium concentration before and after UV flash photolysis of nitrophenyl-EGTA.

**Figure supplement 3.** Expression levels of Munc13-1-EGFP and ubMunc13-2-EGFP.

**Figure supplement 3—source data 1.** Western blots, raw and with bands.

**Figure supplement 3—source data 2.** Quantitative data.

**Figure supplement 4.** PMA-induced recruitment of overexpressed Munc13 proteins to the plasma membrane in WT mouse adrenal chromaffin cells.

**Figure supplement 5.** ubMunc13-2 trafficking in individual chromaffin cells.

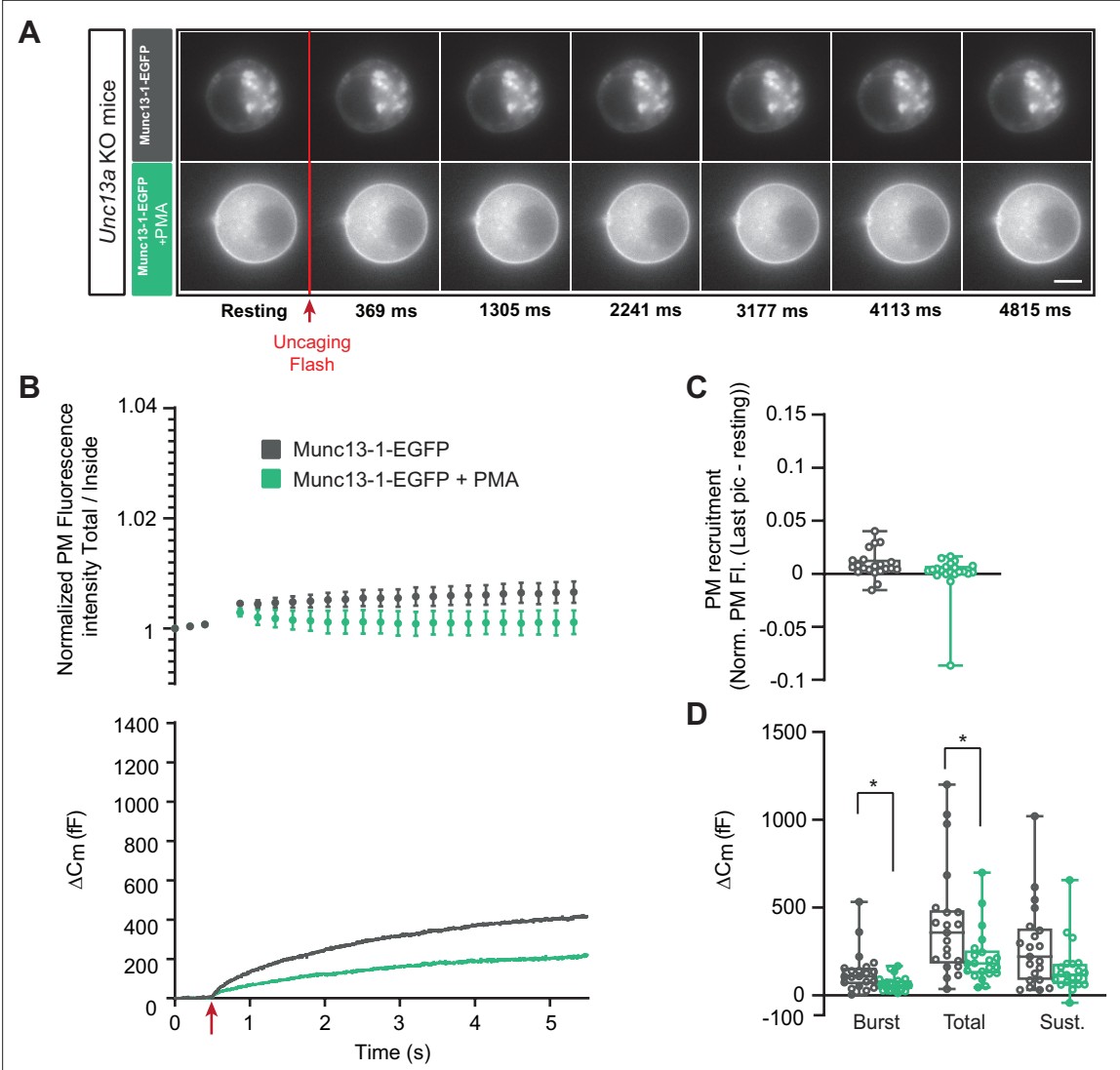

**Figure 4.** Munc13-1-dependent secretion is inhibited by phorbolester. (**A**) Wide-field imaging of Munc13-1-EGFP expressed in *Unc13a* KO chromaffin cells treated with PMA (green) or left untreated (grey). Munc13-1-EGFP was not recruited at the plasma membrane after calcium uncaging (top panel) but was present at the PM upon PMA treatment (bottom panel). The uncaging light flash is represented by a red arrow. Scale bar: 5 µm. (**B**) Top: Quantification of the plasma membrane Munc13-1-EGFP fluorescence intensity normalized to the first picture acquired in resting condition (mean ± SEM of all cells). Bottom: Capacitance traces (mean of all cells) obtained simultaneously with EGFP imaging, showing that PMA treatment reduces secretion in *Unc13a* KO cells overexpressing Munc13-1. (**C**) Plasma membrane Munc13-1-EGFP recruitment (total fluorescence / inside fluorescence) normalized to resting (pre-stimulation) values. (**D**) Sizes of Burst, Total and Sustained release. Data information: data in panel B are presented as mean ± SEM; data in panels C and D are presented as box and whiskers. *: $p<0.05$, Mann-Whitney tests. OE Munc13-1-EGFP: n=21 cells; OE Munc13-1-EGFP+PMA: n=21 cells.

The online version of this article includes the following source data and figure supplement(s) for figure 4:

**Source data 1.** Quantitative data.

**Figure supplement 1.** Munc13-1 overexpression potentiates secretion in WT mouse adrenal chromaffin cells.

**Figure supplement 1—source data 1.** Quantitative data.

**Figure supplement 2.** Munc13-1 trafficking in individual chromaffin cells.

**Figure supplement 3.** PMA treatment of WT mouse adrenal chromaffin cells overexpressing Munc13-1 H567K does not change secretion.

**Figure supplement 3—source data 1.** Quantitative data.

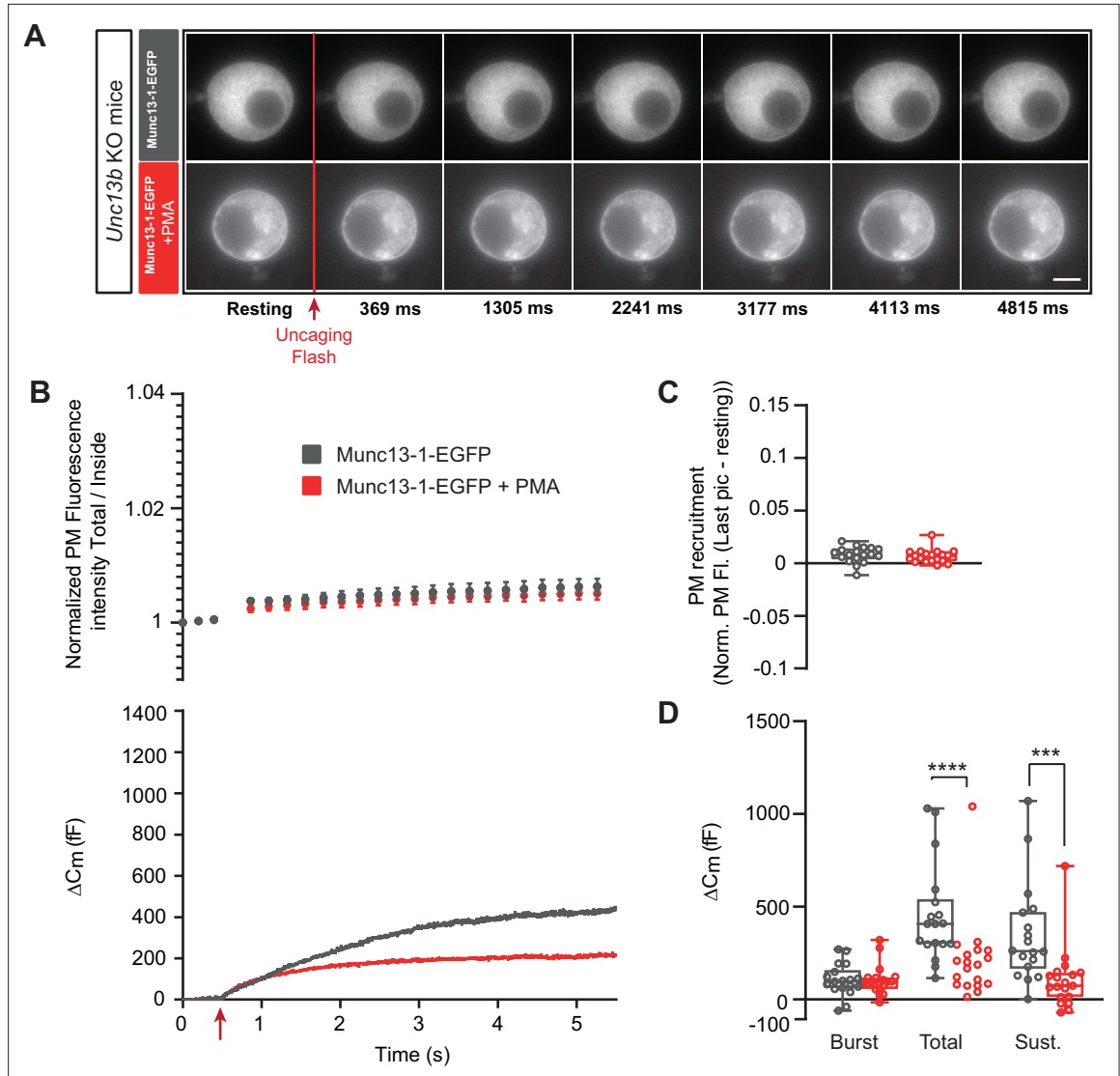

**Figure 5.** Munc13-1-dependent secretion is inhibited by phorbolester in the absence of ubMunc13-2. (**A**) Wide-field imaging of Munc13-1-EGFP expressed in *Unc13b* KO chromaffin cells and treated with PMA (red) or left untreated (grey). Munc13-1-EGFP was not recruited at the plasma membrane after calcium uncaging (top panel) but was present at the PM upon PMA treatment (bottom panel). The uncaging light flash is represented by a red arrow. Scale bar: 5 μm. (**B**) Top: Quantification of the plasma membrane Munc13-1-EGFP fluorescence intensity normalized to the first picture acquired in resting condition (mean ± SEM of all cells). Bottom: Capacitance traces (mean of all cells) obtained simultaneously with EGFP imaging, showing that PMA treatment reduces secretion in *Unc13b* KO cells overexpressing Munc13-1-EGFP. (**C**) Plasma membrane Munc13-1-EGFP recruitment. (**D**) Sizes of Burst, Total and Sustained release. Data information: data in panel B are presented as mean ± SEM; data in panels C and D are presented as box and whiskers. ***: p<0.001; ****: p<0.0001, Mann-Whitney tests. OE Munc13-1-EGFP: n=18 cells; OE Munc13-1-EGFP+PMA: n=19 cells.

The online version of this article includes the following source data for figure 5:

**Source data 1.** Quantitative data.

ubMunc13-2-EGFP expression plasmids (Materials and Methods). Although some individual cells displayed EGFP-Munc13-1 trafficking (*Figure 4—figure supplement 2*), overall little membrane trafficking was induced by increasing calcium in Munc13-1-EGFP expressing cells (*Figure 4B–C*). Following PMA-application, total, burst and sustained secretion were reduced; the effects on total and burst secretion were statistically significant (Burst - OE Munc13-1-EGFP: 136.92 fF ±26.09 fF,+PMA: 68.34 fF ±9.10 fF, Mann-Whitney test: p=0.0117; Total - OE Munc13-1-EGFP: 414.22 fF ±69.47 fF,+PMA: 219.53 fF ±34.35 fF, Mann-Whitney test: p=0.0182; Sustained - OE Munc13-1-EGFP: 273.30 fF ±52.75 fF; 150.9±32.07, Mann-Whitney test: p=0.0793; *Figure 4B and D*). These data indicate that PMA-induced stimulation of Munc13-1 is inhibitory for DV secretion

in chromaffin cells, in agreement with findings above in the *Unc13a* and *Unc13b* mouse lines. As a control, we expressed the Munc13-1 H567K mutation, which does not bind to phorbolesters (*Betz et al., 1998*) in adrenal chromaffin cells from CD1 mice. Upon expression, there was no effect of PMA (*Figure 4—figure supplement 3*), which agrees with previous data from bovine chromaffin cells (*Bauer et al., 2007*).

Thus, evidence from both Munc13-1-EGFP overexpression and *Unc13a* KO as well as *Unc13b* KO indicate that PMA interacting with Munc13-1 is inhibitory, whereas PMA interacting with ubMunc13-2 is positive for DV secretion. Since ubMunc13-2 supports more secretion than Munc13-1 upon overexpression in parallel experiments (*Man et al., 2015*), it is possible that the apparent negative effect of Munc13-1 could be due to competition with ubMunc13-2, because targeting Munc13-1 to the plasma membrane with PMA might displace the more potent ubMunc13-2. To investigate this, we expressed Munc13-1-EGFP in *Unc13b* KO cells (*Figure 5A–D*). Even under these circumstances, PMA application led to a significant decrease in total and sustained secretion, but the secretory burst was now unaffected (Burst - OE Munc13-1-EGFP: 104.64 fF ±20.58 fF,+PMA: 104.47 fF ±18.30 fF, Mann-Whitney test: p=0.8280; Total - OE Munc13-1-EGFP: 452.45 fF ±62.21 fF,+PMA: 210.43 fF ±50.40 fF, Mann-Whitney test: p<0.0001; Sustained - OE Munc13-1-EGFP: 347.8 fF ±63.42 fF;+PMA: 106±38.75, Mann-Whitney test: p=0.0001; *Figure 5D*). Thus, the ability of PMA and Munc13-1 to suppress the burst might be due to displacement of ubMunc13-2, but ubMunc13-2 is not required for the overall inhibitory nature of PMAs interaction with Munc13-1.

## Synaptotagmin-7, ubMunc13-2 and phorbolester form a stimulatory triad for vesicle fusion

We previously showed that Syt7 is involved in calcium and phorbolester-induced priming of chromaffin cells DVs (*Tawfik et al., 2021*). We here set out to understand whether the different abilities of ubMunc13-2 and Munc13-1 to support DV secretion in the presence of PMA depends on syt7.

We first expressed ubMunc13-2-EGFP in *Syt7* KO and WT cells. As previously reported (*Tawfik et al., 2021*), expression of ubMunc13-2 resulted in massive secretion in both the *Syt7* KO and WT, but secretion was delayed in *Syt7* KO (*Figure 6B*). As a result, the burst size, as defined by secretion within the first second, was significantly reduced in *Syt7* KO cells expressing EGFP-ubMunc13-2 compared to *Syt7* WT expressing EGFP-ubMunc13-2 (Burst - OE ubMunc13-2-eGFP in *Syt7* WT: 317.01 fF ±53.24 fF, OE Munc13-2-EGFP in *Syt7* KO: 150.21 fF ±24.91 fF, Mann-Whitney test: p=0.0286, *Figure 6D*; for other values see Source Data File). Imaging showed that in spite of the delayed secretion in the *Syt7* KO, ubMunc13-2-EGFP trafficking was unchanged (*Figure 6A–B*), and – consequently – ubMunc13-2-EGFP trafficking preceded exocytosis in the *Syt7* KO ubMunc13-2 overexpressing cells (*Figure 6B* bottom panel). Therefore, calcium-dependent ubMunc13-2-EGFP trafficking is independent of Syt7. To investigate this further, we performed co-immunoprecipations between b/ubMunc13-2 and Syt7, as well as between Munc13-1 and Syt7 (Materials and methods). However, we failed to identify any interaction (*Figure 6—figure supplement 1*). Therefore, any functional interaction between Syt7 and Munc13-2 does not seem to involve direct stable binding or co-trafficking, but transient interactions cannot be ruled out.

To shed further light on the interaction between PMA, Munc13-2 and Syt7, we applied PMA to *Syt7* KO cells expressing ubMunc13-2-EGFP (*Figure 7*). Strikingly, in these cells, PMA was strongly inhibitory for secretion (*Figure 7D*), leading to a statistically significant decrease in total and sustained secretion, whereas the burst was non-significantly reduced (Burst - OE ubMunc13-2-EGFP in *Syt7* KO: 220.51 fF ±51.60 fF,+PMA: 185.45 fF ±53.19 fF, Mann-Whitney test: p=0.3989; Total - OE ubMunc13-2-EGFP in *Syt7* KO: 896.60 fF ±199.86 fF,+PMA: 223.03 fF ±53.60 fF, Mann-Whitney test: p=0.0002; Sustained - OE ubMunc13-2-EGFP in *Syt7* KO: 676.34 fF ±156.89 fF;+PMA: 37.43±15.53, Mann-Whitney test: p<0.0001; *Figure 7D*). Similar to data obtained above, ubMunc13-2 trafficked to the PM after a calcium increase (*Figure 7A–C*). In a final experiment, we showed that in *Syt7* KO cells expressing Munc13-1, PMA was also inhibitory for both total and sustained secretion (*Figure 7—figure supplement 1*). Overall, the difference between Munc13-1 and ubMunc13-2 in their response to PMA lies in the specific ability of ubMunc13-2 to interact productively with Syt7; in the absence of Syt7, PMA interacting with ubMunc13-2 is negative for secretion, as is the case for Munc13-1.

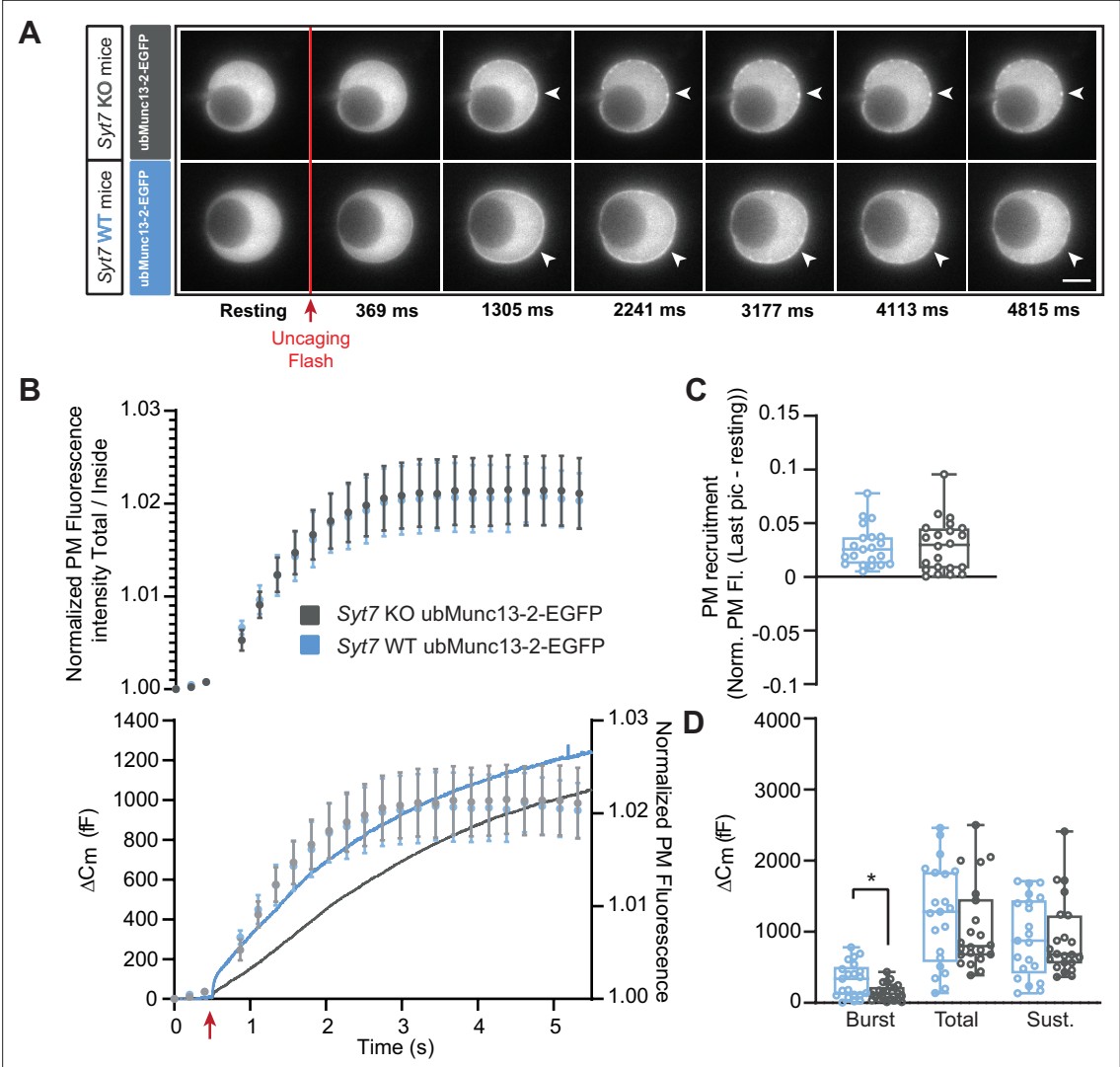

**Figure 6.** Ca²⁺-dependent recruitment of ubMunc13-2 is independent of Synaptotagmin-7. (**A**) Wide-field imaging of ubMunc13-2-EGFP expressed in *Syt7* KO (grey) or *Syt7* WT cells (blue) chromaffin cells. ubMunc13-2-EGFP is recruited to the plasma membrane after calcium uncaging in the presence or absence of Syt7 (white arrows). The uncaging light flash is represented by a red arrow. Scale bar: 5 µm. (**B**) Top: Quantification of the plasma membrane ubMunc13-2-EGFP fluorescence intensity normalized to the first picture acquired in resting condition (mean ± SEM of all cells). Bottom: Capacitance traces (mean of all cells) obtained simultaneously with EGFP imaging show a delay in secretion in *Syt7* KO cells overexpressing ubMunc13-2-EGFP. Right axis: Normalized PM fluorescence replotted from panel (**A**) showing that the time course of ubMunc13-2 precedes the capacitance trace. (**C**) Plasma membrane ubMunc13-2-EGFP recruitment. (**D**) Sizes of the Burst, Total and Sustained release. Data information: data in panel B are presented as mean ± SEM; data in panels C and D are presented as box and whiskers. *: p<0.05, Mann-Whitney test. *Syt7* KO OE ubMunc13-2-EGFP: n=21 cells; *Syt7* WT OE ubMunc13-2-EGFP PMA: n=22 cells.

The online version of this article includes the following source data and figure supplement(s) for figure 6:

**Source data 1.** Quantitative data.

**Figure supplement 1.** Syt-7 is not co-immunoprecipitated with Munc13-2 or Munc13-1.

**Figure supplement 1—source data 1.** Western blots, raw and with bands.

## Discussion

Previous investigations did not resolve the function of Munc13-1 in chromaffin cells, since no consequences of eliminating Munc13-1 were identified (***Man et al., 2015***), while other studies showed that overexpression of Munc13-1 increased secretion in bovine chromaffin cells (***Ashery et al., 2000***; ***Betz et al., 2001***) or mouse *Unc13a/Unc13b* double knockout chromaffin cells (***Man et al., 2015***). Here, we reproduced the positive effect of Munc13-1 overexpression in WT cells (***Figure 4—figure supplement***

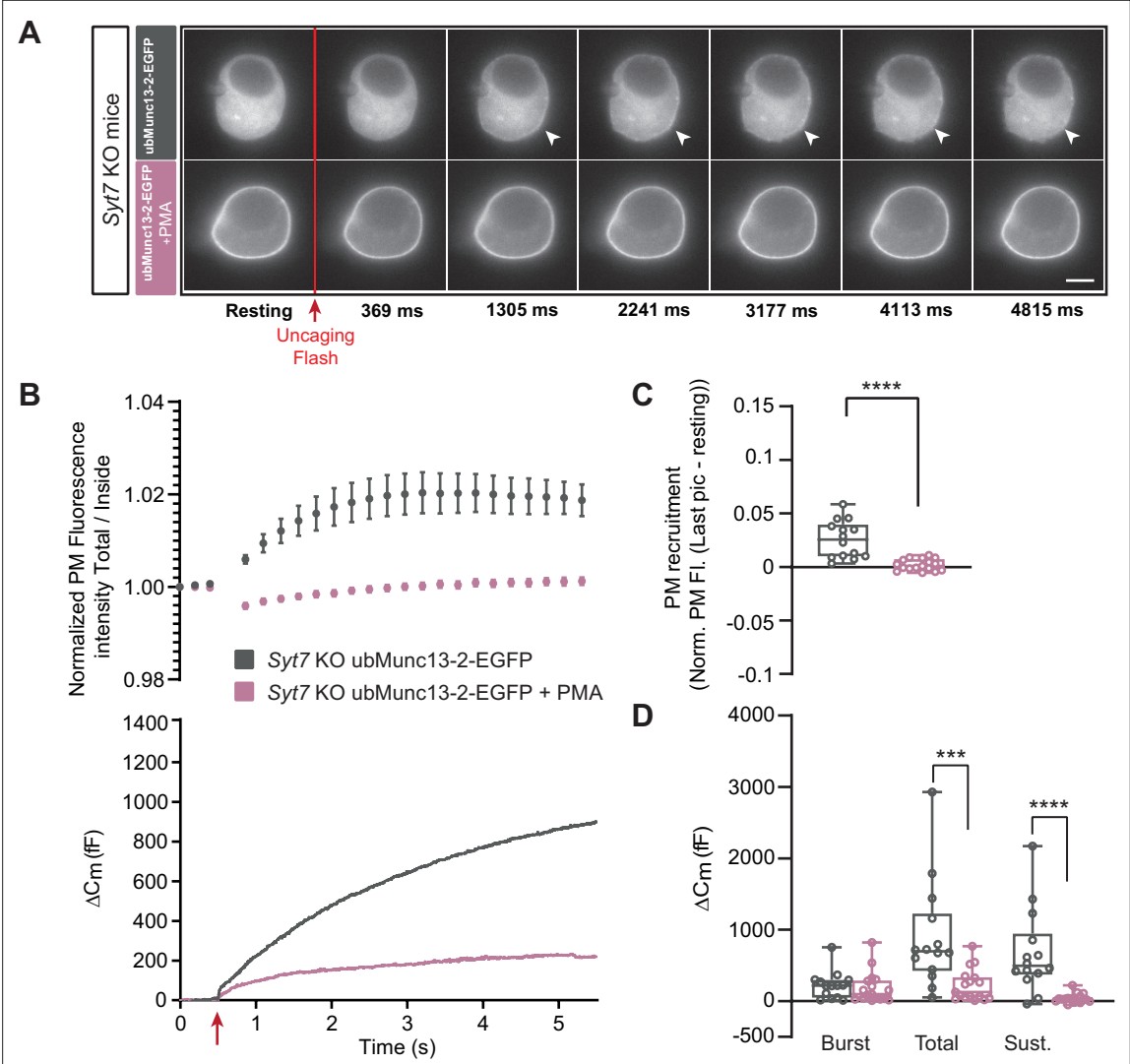

**Figure 7.** In the absence of Synaptotagmin-7, ubMunc13-2-dependent secretion is inhibited by phorbolester. (**A**) Wide-field imaging of overexpressed ubMunc13-2-EGFP expressed in *Syt7* KO chromaffin cells treated with PMA (pink) or left untreated (grey). ubMunc13-2-EGFP was recruited to the plasma membrane after calcium uncaging, and was found at the plasma membrane upon PMA treatment. The uncaging light flash is represented by a red arrow. Scale bar: 5 μm. (**B**) Top: Quantification of the plasma membrane ubMunc13-2-EGFP fluorescence intensity normalized to the first picture acquired in resting condition (mean ± SEM of all cells). Bottom: Capacitance traces (mean of all cells) obtained simultaneously with EGFP imaging in *Syt7* KO cells expressing ubMunc13-2-EGFP in the absence and presence of PMA. Upon exposure to PMA, secretion in ubMunc13-2 expressing *Syt7* *KO* cells was strongly inhibited. (**C**) Plasma membrane ubMunc13-2-EGFP recruitment. (**D**) Sizes of Burst, Total and Sustained release. Data information: data in panel B are presented as mean ± SEM; data in panels C and D are presented as box and whiskers.*: $p<0.05$; ***: $p<0.001$; ****: $p<0.0001$, Mann-Whitney tests. OE ubMunc13-2-EGFP in *Syt7* KO: n=14 cells;+PMA: n=17 cells.

The online version of this article includes the following source data and figure supplement(s) for figure 7:

**Source data 1.** Quantitative data.

**Figure supplement 1.** Munc13-1 inhibits secretion independently of Syt7.

**Figure supplement 1—source data 1.** Quantitative data.

**Figure supplement 2.** PKC inhibition does not affect membrane fusion in the presence or absence of ubMunc13-2.

**Figure supplement 2—source data 1.** Quantitative data.

*1*), but the increase in burst secretion was smaller in our hands than reported earlier upon expression in bovine chromaffin cells (***Ashery et al., 2000***; ***Betz et al., 2001***). This can be due to differences in species or age of the experimental animals (embryonic/newborn mice vs adult cow). It might also be due to differences in prestimulation $Ca^{2+}$ concentrations, which were not measured in each cell in our experiment and

not measured in the work on bovine cells. We also reproduced the lack of consequences of Munc13-1 elimination (*Figure 2*; *Man et al., 2015*). However, we surprisingly found that the dominating Munc13 paralog determines the effect of phorbolester (PMA), such that PMA is inhibitory when Munc13-1 dominates, but stimulatory when ubMunc13-2 is prevalent. The evidence supporting this conclusion is:

1. In the *Unc13b* WT mouse, PMA was stimulatory for secretion, in agreement with the higher endogenous expression level of ubMunc13-2 than Munc13-1 (*Man et al., 2015*). In *Unc13b* KO mouse (expressing endogenous Munc13-1, but no ubMunc13-2), PMA became inhibitory for secretion.
2. In the *Unc13a* KO mouse, PMA was more stimulatory for secretion than in the *Unc13a* WT mouse, consistent with a negative function for endogenous Munc13-1 in the presence of PMA.
3. Upon overexpression of ubMunc13-2, PMA was stimulatory for secretion (as in WT cells).
4. Upon overexpression of Munc13-1, PMA became inhibitory for secretion, whether overexpression was in *Unc13a* KO (still expressing ubMunc13-2) or in *Unc13b* KO cells.

The identification of opposite effects of phorbolester for DV secretion depending on the predominant expression of Munc13-1 and ubMunc13-2 is surprising, given their ability to cross-rescue (*Man et al., 2015*; *Rosenmund et al., 2002*). However, a difference in phorbolester-effect of Munc13-1 and ubMunc13-2 was detected before, as the positive effect of phorbolester on synaptic release was larger in ubMunc13-2 than in Munc13-1 expressing *Unc13a*/*Unc13b* double KO glutamatergic neurons (*Rosenmund et al., 2002*). Nevertheless, phorbolester interacting with Munc13-1 is clearly stimulatory for glutamate release (*Rosenmund et al., 2002*).

The effect of phorbolester on Munc13s is partly to recruit the protein to the plasma membrane by binding to the C1-domain, and partly to release an inhibitory effect of the (DAG and Ca²⁺ unbound) C1-C2B domain (*Michelassi et al., 2017*). Previous findings in chromaffin cells showed that overexpression of Doc2B boosts the secretory burst, which required an intact Munc13-binding domain (the MID domain *Friedrich et al., 2013*; *Michelassi et al., 2017*) and the presence of ubMunc13-2, making it possible for Doc2B to target ubMunc13-2 to the plasma membrane (*Orita et al., 1997*). However, in the absence of ubMunc13-2 (i.e. in *Unc13b* KO cells), overexpression of Doc2B became strongly inhibitory (*Houy et al., 2017*). This inhibitory function of Doc2B aligns with its ability to target Munc13-1 (present in *Unc13b* KO cells) to the plasma membrane (*Friedrich et al., 2013*; *Houy et al., 2017*) and agrees with our findings here that plasma membrane targeting of Munc13-1 by PMA is inhibitory for chromaffin cell DV secretion.

Apart from its ability to bind Munc13s, phorbolester also activates protein kinase C (PKC), which has effects on chromaffin cell excitability, calcium homeostasis, and secretion (*Fulop and Smith, 2006*; *Kuri et al., 2009*; *Park et al., 2006*; *Rosmaninho-Salgado et al., 2007*; *Smith et al., 1998*; *Soma et al., 2009*; *Staal et al., 2008*). Using whole-cell voltage clamp and calcium-uncaging, effects on excitability and calcium homeostasis are effectively bypassed, allowing us to focus on the secretory machinery in the present study. In neurons, the effects of phorbolester have been attributed both to Munc13-1 C1-activation, and PKC-activation (*Rhee et al., 2002*; *Wierda et al., 2007*). PKC phosphorylates exocytotic proteins, including Munc18-1 (*Genc et al., 2014*; *Wierda et al., 2007*) and synaptotagmin-1 (*de Jong et al., 2016*), and these phosphorylation events are required for full potentiation of synaptic transmission after train stimulation or PMA application, respectively (but see *Wang et al., 2021b*). Phosphorylation of Munc18-1 also potentiates secretion from adult bovine chromaffin cells (*Nili et al., 2006*). However, PKC-dependent modulation of the release machinery appears weaker or absent in embryonic or newborn mouse chromaffin cells, since blocking Munc18-1 or synaptotagmin-1 phosphorylation have no detectable consequences, either in control conditions, or after PMA application (*Nagy et al., 2006*; *Nili et al., 2006*). Furthermore, we have shown that blocking PKC-activity has no consequences for capacitance increases during Ca²⁺-uncaging in chromaffin cells, if only one Ca²⁺-stimulation is used (*Nagy et al., 2002*; *Nili et al., 2006*). We confirmed that here for both *Unc13b* WT and KO cells, using a broad spectrum PKC inhibitor (Gö6983, *Figure 7—figure supplement 2*). However, although membrane fusion was unaffected, the amperometric signal was reduced, reflecting a reduced release of adrenaline/noradrenaline – this effect of PKC inhibition was noted before (*Fulop and Smith, 2006*; *Staal et al., 2008*). These observations are in agreement with our finding that Munc13-2 is absolutely required for the stimulatory effect of phorbolester on capacitance increase (*Figure 1*); however, we cannot rule out an interaction between PKC-dependent phosphorylation and Munc13 activation.

Even though overexpression of ubMunc13-2 results in the most potent secretion from chromaffin cells so far known (*Man et al., 2015*; *Zikich et al., 2008*), secretion could be potentiated further with phorbolester (*Figure 3*). However, phorbolester was inhibitory in cells not expressing Syt7 (*Figure 7*). The inhibitory effect depended upon the overexpression of ubMunc13-2, as it was not found in non-overexpressing *Syt7* KO cells (*Tawfik et al., 2021*); indeed, the reduction in secretion by application of PMA brings secretion back to approximately the level in *Syt7* KO cells not overexpressing ubMunc13-2 (*Tawfik et al., 2021*), which indicates that it is the increase in secretion by ubMunc13-2 overexpression which is reversed by PMA in this case. Overall, an interaction between Syt7, ubMunc13-2 and DAG/phorbolester ensures optimal vesicle priming. We previously showed that although phorbolester was ineffective in the absence of Syt7 at low prestimulation [$Ca^{2+}$], at higher prestimulation [$Ca^{2+}$] it regained some potency (*Tawfik et al., 2021*); therefore, in the absence of Syt7 other $Ca^{2+}$-sensors can substitute in the stimulatory 'triad', most likely Syt1 or Doc2B (*Houy et al., 2017*).

We could not detect any effect of Syt7 on $Ca^{2+}$-dependent ubMunc13-2 trafficking, which likely is driven by $Ca^{2+}$-binding to the ubMunc13-2 C2B-domain, or any direct binding in co-immunoprecipitation experiments. Instead, we suggest that the interaction is functional, and a likely explanation is the observation that in the *Syt7* KO, vesicles accumulate at 20–40 nm distance to the plasma membrane (*Tawfik et al., 2021*). This distance is out of range of the rod-shaped elongated structure formed by Munc13 proteins (*Quade et al., 2019*; *Xu et al., 2017*). Recruitment of ubMunc13-2 to the plasma membrane in the absence of Syt7 will therefore not lead to productive bridging of vesicular and plasma membrane, and vesicle priming. Instead, the recruited ubMunc13-2 might inhibit release, by steric hindrance, by adopting an orientation that will not promote membrane bridging (*Camacho et al., 2021*; *Grushin et al., 2022*), or by covering up $PIP_2$-patches in the plasma membrane, which are necessary for vesicle priming and fusion (*Aoyagi et al., 2005*; *James et al., 2008*; *Milosevic et al., 2005*).

Even though phorbolester became inhibitory when Munc13-1 dominated, expression of Munc13-1 did increase secretion in the absence of phorbolester. Munc13-1 is therefore able to stimulate priming or fusion of DVs. Another difference between Munc13-1-EGFP and ubMunc13-2-EGFP was the ability of the latter to traffic to the membrane in parallel with the sustained part of the capacitance trace following an abrupt $Ca^{2+}$ increase. This indicates that ubMunc13-2 recruitment to the plasma membrane might be rate limiting for vesicle priming during sustained secretion. We did not see strong $Ca^{2+}$-dependent recruitment of Munc13-1-EGFP to the membrane (over the course of ~5 s; note that it could recruit over longer times). Phorbolester recruited Munc13-1 to the plasma membrane, but caused inhibition of secretion. Together, these findings suggest that Munc13-1 might not act at the plasma membrane in priming of DVs. Another indication for an action at the vesicular level is the previous observation that the positive effect of Munc13-1-EGFP expression depends on the expression of CAPS proteins, and in CAPS-1/2 DKO chromaffin cells overexpression of Munc13-1-EGFP inhibited secretion (*Liu et al., 2010*). CAPS is a vesicle-associated protein (*Ann et al., 1997*; *Kabachinski et al., 2016*), which contains a SNARE-binding MUN-domain (as Munc13s) and a PIP2-binding PH domain. The PH-domain is essential for its action in vesicle exocytosis (*Grishanin et al., 2004*; *James et al., 2010*; *Nguyen Truong et al., 2014*), as it directs vesicles to dock and prime at $PIP_2$-sites on the plasma membrane (*Kabachinski et al., 2016*). CAPS and Munc13 proteins do not cross-rescue (*Jockusch et al., 2007*; *Kabachinski et al., 2014*), but ubMunc13-2 can overcome the need for CAPS when DAG is abundant, due to binding to ubMunc13-2's C1-domain (*Kabachinski et al., 2014*). This effect is likely related to the stimulatory effect of phorbolester in the presence of ubMunc13-2 that we identified here.

The negative effect of Munc13-1 in the presence of phorbolester could be caused by recruitment of Munc13-1 to the plasma membrane, which takes it away from its CAPS-dependent role, or it could be caused by the displacement of ubMunc13-2 at the plasma membrane. Both effects are apparently involved, since expression of Munc13-1-EGFP in the presence of endogenous ubMunc13-2 inhibited the burst (*Figure 4*), which was not the case when Munc13-1-EGFP was expressed in *Unc13b* KO cells (*Figure 5*). Nevertheless, the data also indicate that there is a negative effect of PMA x Munc13-1, which is independent of Munc13-2.

Our data seem at odds with previous finding that expressing the Munc13-1 DAG-insensitive H567K mutant attenuated the effect of histamine in bovine chromaffin cells (*Bauer et al., 2007*). Histamine acts on $G_q$-coupled H1-receptors to increase DAG levels in chromaffin cells. However, the

observations are probably consistent, because overexpressing the Munc13-1 H567K-mutation might overwhelm endogenous ubMunc13-2. Indeed, upon overexpression of Munc13-1 H567K, phorbolester was without effect (*Figure 4—figure supplement 3*).

Given the very similar domain structure of Munc13-1 and ubMunc13-2, it is surprising to find different functions of the two proteins in DV fusion. However, it has already been reported that the C1-C2-MUN domain of Munc13-1 stimulates SV fusion, but not DV fusion in an in vitro assay (*Kreutzberger et al., 2017*; *Kreutzberger et al., 2019*). Several possibilities might explain these findings. Munc13-1 is a slightly longer protein that harbors an additional stretch of amino acids with unknown function immediately N-terminal of the CaM-binding site (*Lipstein et al., 2012*). This stretch might confer preferential interaction with SVs over DVs, or it might interact with the membrane. Another possibility is that there might be subtle differences in the length, shape or angle with the PM of the rod-like domain in Munc13-1 and ubMunc13-2, which bridges vesicular and plasma membrane (*Padmanarayana et al., 2021*; *Quade et al., 2019*; *Xu et al., 2017*). Strikingly, there is evidence that Munc13-1 can adopt different angles on the plasma membrane (*Grushin et al., 2022*) and that the predominant angle changes upon binding to DAG via its C1-domain (*Camacho et al., 2021*). For priming to occur, the angle might need to accurately fit the curvature of the vesicle, which is markedly different for SVs and DVs. A difference in the angle formed by DAG-bound Munc13-1 and ubMunc13-2 could confer preferential function towards SVs or DVs, respectively. A difference in the ability to oligomerize between vesicular membrane and PM might also be involved (*Grushin et al., 2022*). However, until now comparative structural studies of ubMunc13-2 and Munc13-1 that could help solve this question are missing.

Overall, we found two striking differences between Munc13-1 and ubMunc13-2 in chromaffin cells: only ubMunc13-2 traffics to the plasma membrane with a time course matching secretion after an $[Ca^{2+}]$ increase, and only ubMunc13-2 supports more secretion in the presence of phorbolester. The latter effect depends on endogenous Syt7 expression, indicating that Munc13-2, Syt7 and DAG/phorbolesters form a stimulatory triad for DV fusion. As a consequence of the inability of Munc13-1 to substitute for ubMunc13-2 action, the ubMunc13-2/Munc13-1 ratio might determine the effect of DAG/phorbolesters/ in chromaffin cells.

# Materials and methods

## Key resources table

| Reagent type (species) or resource | Designation | Source or reference | Identifiers | Additional information |
|---|---|---|---|---|
| Strain, strain background (*M. musculus*) | CD1 | Experimental Medicine, Panum Stable, University of Copenhagen. | | |
| Genetic reagent (*M. musculus*) | *Synaptotagmin-7 (Syt7)* null allele | *Maximov et al., 2008* | PMID:18308933 | |
| Genetic reagent (*M. musculus*) | *Unc13a* null allele | *Augustin et al., 1999a* | PMID:10440375 | |
| Genetic reagent (*M. musculus*) | *Unc13b* null allele | *Varoqueaux et al., 2005* | PMID:12070347 | |
| Transfected construct (*Rattus Norvegicus*) | pSFV1-ubmunc13-2-EGFP | *Zikich et al., 2008* | PMID:18287511 Local identifier: 486 | Gift from Sonja Wojcik |
| Transfected construct (*Rattus Norvegicus*) | pSFV1-ubmunc13-1-EGFP | *Ashery et al., 1999* | PMID:10494858 Local identifier: 487 | Gift from Sonja Wojcik |
| Antibody | anti-Chromogranin A (Rabbit polyclonal) | Abcam | Ab15160 RRID: AB_301704 | 1:500; Overnight at 4 degrees |
| Antibody | anti-TH (Mouse monoclonal) | Merck Millipore | MAB318 RRID: AB_2201528 | 1:2000; Overnight at 4 degrees |
| Antibody | anti-Synaptotagmin-7 (Rabbit polyclonal) | Synaptic System | SYSY: 105173 RRID: AB_887838 | 1:500; Overnight at 4 degrees. |

*Continued on next page*

*Continued*

| Reagent type (species) or resource | Designation | Source or reference | Identifiers | Additional information |
|---|---|---|---|---|
| Antibody | Anti-Munc13-1 (mouse monoclonal) | Home-made (Max Planck Institute of Multidisciplinary Sciences, Göttingen) *Betz et al., 1998* | PMID:9697857 | 1:1000 1 hr at room temperature |
| Antibody | Anti-Munc13-1 (Rabbit polyclonal) | Home-made (Max Planck Institute of Multidisciplinary Sciences, Göttingen) *Varoqueaux et al., 2005* | PMID:15988013 | Co-IP: 1:300 1 hr at 4 degrees |
| Antibody | anti-bMunc13 (Rabbit polyclonal) | Synaptic System | SYSY126203 RRID: AB_2619807 | 1:1000; Overnight at 4 degrees |
| Antibody | Anti-β-Actin−Peroxidase antibody (Mouse monoclonal) | Sigma-aldrich | A3854 RRID:AB_262011 | 1:10000; 30 min at room temperature |
| Antibody | anti-rabbit HRP (Goat polyclonal) | Agilent | Dako-P0448 RRID: AB_2617138 | 1:2000; 1 hr and 30 min at room temperature |
| Antibody | anti-rabbit Alexa 488 (Goat polyclonal) | ThermoFisher Scientific | A11008 RRID: AB_143165 | 1:500; 30 min at room temperature |
| Antibody | anti-mouse Alexa 647 (Goat polyclonal) | ThermoFisher Scientific | A21235 RRID: AB_2535804 | 1:500; 30 min at room temperature |
| Antibody | anti-GFP (Rabbit polyclonal) | Synaptic Systems | SYSY132003 RRID: AB_1834147 | 1:300; Overnight at 4 degrees |
| Commercial assay or kit | BCA Protein assay kit | Pierce | Pierce: 23227 | |
| Commercial assay or kit | GFP-Trap | Chromotek | gtak-20 | |
| Chemical compound, drug | NaCl | Sigma-aldrich | Sigma-aldrich: S9888 | |
| Chemical compound, drug | KCl | Sigma-aldrich | Sigma-aldrich: P5405 | |
| Chemical compound, drug | NaH2PO4 | Sigma-aldrich | Sigma-aldrich: S8282 | |
| Chemical compound, drug | Glucose | Sigma-aldrich | Sigma-aldrich: G8270 | |
| Chemical compound, drug | DMEM | Gibco | Gibco: 31966047 | |
| Chemical compound, drug | L-cysteine | Sigma-aldrich | Sigma-aldrich: C7352 | |
| Chemical compound, drug | CaCl2 | Sigma-aldrich | Sigma-aldrich: 499609 | |
| Chemical compound, drug | EDTA | Sigma-aldrich | Sigma-aldrich: E5134 | |
| Chemical compound, drug | papain | Worthington Biochemical | Worthington Biochemical: LS003126 | |
| Chemical compound, drug | albumin | Sigma-aldrich | Sigma-aldrich: A3095 | |
| Chemical compound, drug | trypsin-inhibitor | Sigma-aldrich | Sigma-aldrich: T9253 | |
| Chemical compound, drug | penicillin/ streptomycin | Invitrogen | Invitrogen: 15140122 | |
| Chemical compound, drug | insulin-transferrin-selenium-X | Invitrogen | Invitrogen: 51500056 | |
| Chemical compound, drug | fetal calf serum | Invitrogen | Invitrogen: 10500064 | |
| Chemical compound, drug | MgCl2 | Sigma-aldrich | Sigma-aldrich: 449172 | |
| Chemical compound, drug | HEPES | Sigma-aldrich | Sigma-aldrich: H3375 | |
| Chemical compound, drug | Nitrophenyl-EGTA (NPE) | Synthesized at the Max-Planck-Institute for biophysical chemistry, Göttingen. | | |
| Chemical compound, drug | Fura-4F | Invitrogen | Invitrogen: F14174 | |
| Chemical compound, drug | Furaptra | Invitrogen | Invitrogen: M1290 | |

*Continued on next page*

*Continued*

| Reagent type (species) or resource | Designation | Source or reference | Identifiers | Additional information |
|---|---|---|---|---|
| Chemical compound, drug | Mg-ATP | Sigma-aldrich | Sigma-aldrich: A9187 | |
| Chemical compound, drug | GTP | Sigma-aldrich | Sigma-aldrich: G8877 | |
| Chemical compound, drug | Vitamin C | Sigma-aldrich | Sigma-aldrich: A5960 | |
| Chemical compound, drug | EGTA | Sigma-aldrich | Sigma-aldrich: E4378 | |
| Chemical compound, drug | Paraformaldehyde | Sigma-aldrich | Sigma-aldrich: P6148 | |
| Chemical compound, drug | PIPES | Sigma-aldrich | Sigma-aldrich: 80635 | |
| Chemical compound, drug | Triton X-100 | Sigma-aldrich | Sigma-aldrich: T8787 | |
| Chemical compound, drug | BSA | Sigma-aldrich | Sigma-aldrich: A4503 | |
| Chemical compound, drug | Prolong Gold | Invitrogen | Invitrogen: P36934 | |
| Chemical compound, drug | Protease cocktail inhibitor | Invitrogen | Invitrogen: 87785 | |
| Chemical compound, drug | RIPA buffer | Invitrogen | Invitrogen: R0278 | |
| Chemical compound, drug | ECL plus western blotting substrate | Pierce | Pierce: 32132 | |
| Chemical compound, drug | Go 6983 | Tocris | Tocris:133053-19-7 PubChem ID: 3499 | |
| Software, algorithm | Igor | wavemetrics | Versions 6.2.1.0 and 8.0.4.2 | |
| Software, algorithm | ImageJ | NIH software | Version 1.53e | |

## Mouse lines and cell culture

Mouse lines, C57/Bl6 *Unc13a* (*Augustin et al., 1999b*), C57/Bl6 *Unc13b* (*Varoqueaux et al., 2002*), C57/Bl6 *Syt7* (*Maximov et al., 2008*), were kept in an AAALAC-accredited stable at the University of Copenhagen operating a 12 hr/12 hr light/dark cycle with access to water and food ad libitum. Permission to keep and breed KO mice was obtained from the Danish Animal Experiments Inspectorate (permission 2018-15-0202-00157). *Unc13b* and *Syt7* knockout (KO) and WT cells were obtained from P0-P2 pups of either sex originating from heterozygous crossing and identified by PCR genotyping. Because *Unc13a* KO animals die within a few hours of birth (*Augustin et al., 1999b*), we obtained *Unc13a* cells from embryos of either sex at embryonic day 18 (E18) from crosses of *Unc13a* heterozygous (+/-) mice. *Unc13a* KO and WT animals were identified by PCR genotyping. Some of the experiments made use of wild-type chromaffin cells from a CD1 outbred mouse strain. Note that *Unc13a* WT, *Unc13b* WT, *Syt7* WT and CD1 WT cells are all wild type, but originate from different mouse lines. Comparison of a KO was always made to WT littermates. Adrenal chromaffin cells primary culture was described previously (*Houy et al., 2021*; *Sørensen et al., 2003*; *Tawfik et al., 2021*). Briefly, the adrenal glands were dissected out and cleaned in Locke's solution (mM: 154 NaCl, 5.6 KCl, 0.85 $NaH_2PO_4$, 2.15 $Na_2HPO_4$, and 10 glucose; and adjusted to pH 7.0). The glands were digested with papain enzyme (20–25 units/ml) for 45 min at 37 °C and 8% $CO_2$ followed by 10–15 min inactivation with DMEM-Inactivation solution. Cells were dissociated and plated in a drop of medium on glass coverslips for 30–45 min, and finally supplemented with 1–2 ml of enriched DMEM media. Cells were used 2–5 days after plating.

## Viral constructs

Acute expression of EGFP-fused ubMunc13-2, Munc13-1 or Munc13-1 H567K was induced by infection with Semliki Forest Virus constructs (*Ashery et al., 2000*; *Zikich et al., 2008*). The constructs (a gift from Nils Brose and Sonja Wojcik, Göttingen) were verified by sequencing with a specific set of primers for each construct. For Munc13-1-EGFP and Munc13-1 H567K-EGFP, the following primers were used; 5´-ATC CCAATTCGGCACGAGC-3´; 5´-CCG CCT TAC TAC ACG ACT TC-3´; 5´-CAC CTG AAG AGA AGG CAG CT-3´; 5´-GCC TGA GAT CTT CGA GCT G-3´; 5´-CGA CGC CTG GAA GGT TTA C-3´; 5´-TCC TAC ACA CCC TGC CTC A-3´; 5´-GAA CCC TGA GGG AGC TGC A-3´; 5´-CCA CCG ACC TGC TCA TCA AA-3´; 5´-TCA AGT CCG ACA CGC GCT-3´; 5´-TCA ATT AAT TAC CCG GCC GC-3´; 5´-CGC ATT TAC GGC GCT GAT GA-3´. The resulting sequences differ from the

primary source sequence available at the Rat Genome Database (RGD ID: 619722) by two substitutions, L756W and E1666G. These two apparent substitutions are likely caused by sequencing errors in the original constructs (*Brose et al., 1995*), since the residues are W and G, respectively, in the Munc13-2 and Munc13-3 published in the same paper, and in *C. elegans unc-13* (*Maruyama and Brenner, 1991*)(Nils Brose, personal communication). The construct furthermore contains two deletions of 23 and 19 amino acids, which correspond to alternatively spliced exons (*Brose et al., 1995*). All previous work is based on Munc13-1 versions lacking these exons (Nils Brose, personal communication).

The following primers were used to sequence Munc13-2-EGFP: 5′-ACA CCT CTA CGG CGG TCC TAG-3′; 5′-GAA AGC CAG AGG AAG CTG TT-3′; 5′AGG TGA ATC CAA GGA GAG AGA T-3′; 5′-AGA CCT GCT CAA TGC CGA TTG-3′; 5′-AGG GGC TAT CCG ACT GCA AAT-3′; 5′-AGG GGC TAT CCG ACT GCA A-3′; 5′-ACA GTG GAC TTG CTG ACC AG-3′; 5′-GAC GTG TCC CTG GAA TTC CT-3′; 5′-GAT TTC CTG GAT GGC AAC CTC-3′; 5′-CTG GCA CTG GGG AGC ATA A-3′; 5′-AGT GGC CTG CAA CGG CCA AGG AC-3′. The obtained sequencing results were in alignment with the primary source sequence available at the Rat Genome Database (RGD ID: 619723).

Semliki Forest Virus particles were produced as previously described (*Ashery et al., 1999*), and activated using chymotrypsin for 45 min, following by inactivation with aprotinin. Measurements were carried out 12–17 hr after transfection of chromaffin cells.

## Immunostaining and confocal microscopy

Cells were plated on 25 mg/ml poly-D-lysine (Sigma P7405) coated coverslips and fixated with 4% Paraformaldehyde (PFA; EMC 15710) and 0.2% Glutaraldehyde (Merck Millipore 104239) for 15 min at room temperature (RT), followed by 2% PFA for an additional 10 min at room temperature (RT). Cells were permeabilized with 0.15% Triton-X100 (Sigma-Aldrich T8787) for 15 min at RT and subsequently blocked with 0.2% cold fish gelatin (Sigma-Aldrich G7765), 1% goat serum (Thermo Fisher Scientific 16210064) and 3% Bovine Albumin Serum (Sigma-Aldrich A4503) for 1 hr at RT. Cells were washed with PBS and glutaraldehyde autofluorescence was quenched with 0.1% Sodium Borohydride (NaBH4; Sigma-Aldrich 213462). Primary antibodies were diluted in blocking solution and incubated as follow: rabbit polyclonal α-CgA (1:500; Abcam 15160) and mouse monoclonal α-TH (1:2000; Merck Millipore MAB318) overnight at 4 °C. Secondary antibodies used were goat α-rabbit Alexa Fluor 488 conjugate (1:500, Abcam ab150169), goat α-mouse Alexa Fluor 647 conjugate (1:500, Thermo Fisher Scientific A21235). Immunofluorescence was visualized using a Zeiss LSM 780 inverted confocal with oil-immersion Plan-Apochromat NA 1.4 63 x objective. The fluorophores were excited with Argon 488 nm (25 mW) and HeNe 633 nm (5 mW) lasers. For each cell, an image stack was obtained, and quantification of CgA was performed using ImageJ software (version 1.53e) on the image plane where the diameter of the cell was largest. Integrated densities of ROIs manually drawn around the entire cell (Total intensity) or excluding the rim of the cell (Inside intensity) were background subtracted. The Total intensity divided by the Inside intensity was used as a measure for plasma membrane localization (*Figure 1O*). *Unc13b* WT and KO cells were acquired on the same day, and laser power, gain and emission detection were unchanged.

## Western blot

Extracts from HEK293FT cells expressing ubMunc13-2-EGFP or Munc13-1-EGFP, as well as non-expressing cells, were collected and lysed in RIPA buffer supplemented with Protease Inhibitor Cocktail (Invitrogen, 89900). The supernatants were collected and protein concentrations were estimated using the BCA Protein Assay Kit (Pierce 23227) after plotting the resulting BSA curve. 25 mg of protein was resolved by 4–12% SDS-PAGE (Invitrogen, Thermo Fisher Scientific) and wet-transferred onto an Amersham Hybond LFP PVDF membrane (GE Healthcare). The membrane was blotted with rabbit polyclonal α-GFP (1:300; Synaptic Systems SY132003) and HRP-conjugated mouse monoclonal α-β-Actin-Peroxidase (1:10000; Sigma-Aldrich A3854), as a loading control, followed by HRP-conjugated α-rabbit (1:2000; Agilent Dako-P0448) secondary antibody. The blot was developed by chemiluminescence Pierce ECL Plus Western blotting substrate systems (Thermofisher Scientific) and immunoreactive bands were detected using the FluorChemE image acquisition system (Protein-Simple) equipped with a cooled CCD camera.

## Immunoprecipitation experiments

Immunoprecipitation of Munc13-2: Brain tissue from the Munc13-2-EYFP knockin mouse (*Cooper et al., 2012*) (a gift from Nils Brose, Max-Planck Institute for experimental medicine, Göttingen) was lysed and homogenized in 10 mM Tris/Cl pH 7.5; 150 mM NaCl; 0.5 mM EDTA; 0.5% NP-40 lysis buffer supplemented with Protease Inhibitor Cocktail (Invitrogen, 89900), with the aid of an homogenizer. Pairs of samples of protein extracts used in the immunoprecipitation (input), supernatant (non-bound) and eluted supernatant were obtained by the protocol provided by GFP-Trap_A Chromotek. One of the samples from each pair was incubated with 1 mM $Ca^{2+}$. Samples were resolved by 4–12% SDS-PAGE (Invitrogen, Thermofisher Scientific) and wet-transferred onto an Amershan Hybond LFP PVDF membrane (GE Healthcare). The membrane was blotted with rabbit polyclonal α-Syt7 (1:500; Synaptic Systems SY105173) and rabbit polyclonal α-bMunc13-2 (1:1000; Synaptic Systems SY126203), followed by HRP-conjugated α-rabbit (1:2000; Agilent Dako-P0448) secondary antibody. The blot was revealed by chemiluminescence Pierce ECL Plus Western blotting substrate system (Thermofisher Scientific) and the bands detected using the FluorChemE image acquisition system (ProteinSimple) equipped with a cooled CCD camera.

Immunoprecipitation of Munc13-1: Cerebral corti of adult (8–11 weeks) WT mice were used to prepare a P2 synaptosomal fraction. The fraction was solubilized in a buffer containing 50 mM Tris/HCl pH 8, 150 mM NaCl, 1 mM CaCl2, 1 mM EGTA, 1% IGEPAL, 0.2 mM phenylmethylsulfonyl fluoride, 1 mg/ml aprotinin, and 0.5 mg/ml leupeptin (solubilization buffer), to a final protein concentration of 2 mg/ml, and ultracentrifuged at 100,000 g to remove insoluble material. A sample ('input') was collected, and the remaining fraction was incubated for 1 hr with a home-made mouse monoclonal-anti-Munc13-1 antibody (3H5 *Betz et al., 1998*) in solubilization buffer. Next, Sepharose-Protein G beads (Invitrogen) were added to capture the antibody and associated proteins, and incubated in rotation for 1 hr. The samples were washed five times with solubilization buffer containing 0.1% IGEPAL to remove background, and eluted using denaturing Laemmli buffer ('IP' samples). Western blot analysis was performed on 4–12% gradient Bis-Tris polyacrylamide gels (Invitrogen) using a home-made rabbit polyclonal antibody against Munc13-1 (*Varoqueaux et al., 2005*) and a rabbit polyclonal antibody (105 173, Synaptic Systems) against Syt7.

## Electrophysiology

Two different electrophysiological setups have been used in this study: 1. The first one combining capacitance measurements, carbon-fiber amperometry, calcium uncaging and calcium concentration detection (Setup 1, used for *Figures 1 and 2*, and *Figure 7—figure supplement 2*), and 2. the second one combining capacitance measurement, calcium uncaging and GFP imaging (Setup 2; *Figure 3—figure supplement 1*, used for all other data).

In setup 1, exocytosis was monitored by combining membrane capacitance measurements and carbon fibre amperometry (*Houy et al., 2021*). Capacitance measurements were based on the Lindau-Neher technique using Pulse HEKA software with Lock-In extension. A 70 mV peak-to-peak sinusoid (1000 Hz) was applied around a holding potential of –70 mV in the whole-cell configuration. The clamp currents were filtered at 3 kHz and recorded at 12 kHz with an EPC9 HEKA amplifier. Secretion was triggered by 1–2ms UV flash-photolysis of the caged $Ca^{2+}$ compound nitrophenyl-EGTA, infused through the patch pipette. The UV-flash delivered from a flash lamp (Rapp Optoelectronic, JML-C2) was bandpass-filtered around 395 nm, transmitted through a light guide and a dual condenser and focused with a Fluor 40 X/N.A. 1.30 oil objective.

The intracellular $Ca^{2+}$ concentration was determined as described in *Nagy et al., 2002*. Two florescent dyes with different affinities toward $Ca^{2+}$, Fura4F (Kd = 1 µM) and furaptra (Kd = 40 µM) were infused via the pipette into the cell. For ratiometric detection, alternating monochromator excitations of 350 nm and 380 nm were generated at 40 Hz and emission was detected via a photodiode, recorded at 3 kHz and filtered at 12 kHz. The 350/380 ratio was pre-calibrated by infusing the cell with known $Ca^{2+}$ concentrations. Amperometric recordings were performed as previously described (*Bruns, 2004*) using a carbon fibre (5–10 µm diameter) insulated with polyethylene and mounted in glass pipette. The fibre was clamped at +700 mV, currents were filtered at 5 kHz and sampled at 25 kHz by an EPC7 HEKA amplifier. Kinetic analysis was performed with Igor Pro software (Wavemetrics – Version 8.04) using a semi-automatic procedure, as previously described (*Tawfik et al., 2021*).

For capacitance measurements in conjunction with EGFP-imaging (setup 2), capacitance measurements and calcium-uncaging were performed with the same method as described above and data were processed in Igor Pro software (Wavemetrics – version 6.21). Burst secretion (approximately corresponding to the sum of the RRP and SRP sizes) was measured at 0.5 s after the flash and the sustained release was obtained by subtracting the total release (capacitance amplitude 5 s after the flash) by the amplitude of the burst. For GFP imaging, image acquisition was performed using a CCD camera (SensiCam, pco.imaging). The protocol of picture acquisitions was the following: 3 pictures in resting condition (202ms interval- 200ms exposure time) followed by the UV-flash photolysis of caged calcium (100ms after the last Resting picture), followed by the acquisition of 20 pictures every 502ms and starting 369ms after the UV flash.

Quantification of the GFP signal was performed in ImageJ software (version 1.53e). Integrated densities of ROIs manually drawn around the entire cell (Total intensity) or excluding the rim of the cell (Inside intensity) were background subtracted. The Total intensity divided by the Inside intensity was used as a measure for plasma membrane localization. Data from all conditions were collected in parallel, using the same culture(s) on the same day(s).

To characterize the PMA-induced recruitment kinetic of Munc13 proteins, 100 nM PMA was added on mouse adrenal chromaffin cells (setup 2) expressing either ubMunc13-2-EGFP, Munc13-1-EGFP or Munc13-1-H567K-EGFP. For GFP-imaging, the image acquisition protocol was the following: 1 picture every 30 s from 30 s to 8 min after adding the PMA followed by 1 picture per min from 8 to 12 min.

The pipette solution contained (in mM): 100 Cs-glutamate, 8 NaCl, 4 $CaCl_2$, 32 Cs-HEPES, 2 Mg-ATP, 0.3 GTP, 5 NPE, 0.4 fura-4F, 0.4 furaptra, and 1 vitamin C. Adjusted to pH 7.2 and osmolarity to 295 mOsm. The extracellular solution contained (in mM): 145 NaCl, 2.8 KCl, 2 $CaCl_2$, 1 $MgCl_2$, 10 HEPES, and 11 glucose. Adjusted to pH 7.2 and osmolarity to 305 mOsm. Phorbol 12-myristate 13-acetate (PMA) (Sigma P8139) and the PKC inhibitor Gö6983 (Tocris 2285) were dissolved in DMSO and diluted in extracellular solution immediately prior to the experiment, to a final concentration of 100 nM and 500 nM, respectively, and used within an hour.

## Statistics

The data are presented as mean and SEM in the text; median values are given in the source data files; n indicates the number of cells and N the number of cell preparations. The parameters estimated here sometimes fulfills the requirements for parametric testing, and sometimes not. To ensure a uniform method of statistical testing, we used non-parametric tests, unless otherwise noted. Non-parametric Mann-Whitney test was used to test changes between two experimental groups, whereas Kruskal-Wallis with post Dunn's test were applied when more than two groups were compared.

## Acknowledgements

We thank Dorte Lauritsen for excellent technical assistance, including genotyping of animals. This investigation was supported by the Novo Nordic Foundation (NNF19OC0058298, JBS), the Independent Research Fund Denmark (0134–00141 A, JBS), the Lundbeck Foundation (R277-2018-802, JBS) and the Deutsche Forschungsgemeinschaft (German Research Foundation, DFG; EXC-2049–390688087 and SFB1286/A11, NL). We are thankful to Nils Brose and Sonja Wojcik, Max-Planck-Institute for Multidisciplinary Sciences, Göttingen, for providing knockout mice, EGFP-fused Munc13-1, Munc13-1 H567K and ubMunc13-2 SFV plasmids, and brain tissue from the EYFP-Munc13-2 mouse.

## Additional information

### Funding

| Funder | Grant reference number | Author |
| --- | --- | --- |
| Novo Nordisk Fonden | NNF19OC0058298 | Jakob Balslev Sørensen |
| Independent Research Fund Denmark | 0134-00141A | Jakob Balslev Sørensen |
| Lundbeckfonden | R277-2018-802 | Jakob Balslev Sørensen |

| Funder | Grant reference number | Author |
|---|---|---|
| Deutsche Forschungsgemeinschaft | EXC-2049 - 390688087 | Noa Lipstein |
| Deutsche Forschungsgemeinschaft | SFB1286/A11 | Noa Lipstein |

The funders had no role in study design, data collection and interpretation, or the decision to submit the work for publication.

## Author contributions

Sébastien Houy, Joana S Martins, Conceptualization, Formal analysis, Investigation, Writing – review and editing; Noa Lipstein, Investigation, Writing – review and editing; Jakob Balslev Sørensen, Conceptualization, Supervision, Funding acquisition, Writing - original draft, Project administration

## Author ORCIDs

Joana S Martins ⓘ http://orcid.org/0000-0002-6721-2935
Noa Lipstein ⓘ http://orcid.org/0000-0002-0755-5899
Jakob Balslev Sørensen ⓘ http://orcid.org/0000-0001-5465-3769

## Decision letter and Author response

Decision letter https://doi.org/10.7554/eLife.79433.sa1
Author response https://doi.org/10.7554/eLife.79433.sa2

---

# Additional files

## Supplementary files

• MDAR checklist

## Data availability

All data generates or analysed during this study are included in the manuscript and supporting files; the source data files contain the numerical data used to generate the figures.

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
