## [Editor Report]

This fundamental study study reveals that phorbol esters have a stimulatory effect on chromaffin cell secretion via ubMunc13-2 but an inhibitory effect via Munc13-1. These convincingly demonstrated, opposing effects of the two closely related Munc13 paralogs are surprising and, although it remains unclear how these findings relate to the mechanism of synaptic vesicle release, the study reveals important differences between the two isoforms of this central priming protein.

---

## [Decision Letter]

**Decision letter after peer review:**

Thank you for submitting your article "Phorbolester-activated Munc13-1 and ubMunc13-2 exert opposing effects on dense-core vesicle secretion" for consideration by *eLife*. Your article has been reviewed by 3 peer reviewers, and the evaluation has been overseen by Axel Brunger as Reviewing Editor and Suzanne Pfeffer as the Senior Editor. The reviewers have opted to remain anonymous.

Essential revisions:

1. The conserved domain architectures of Munc13-1 and ubMun13-2 are remarkably similar. Thus, the results from this study are surprising. Please comment and speculate how the different, but closely related Munc13 paralogs can have such different mechanisms.

2. It is claimed that " plasma membrane targeting of Munc13-1 is inhibitory for chromaffin cell DV secretion" (line 498). However, overexpression of Munc13-1 facilitates the release in wild-type cells (Figure 4-S1). Should one expect this overexpressed Munc13-1 not to increase the Ca^2+^-dependent membrane targeting?

3. The authors noted that in the Rosenmund et al. 2002 paper, they had shown that phorbol ester potentiated glutamate release (by more than twofold). Ideally, please perform an experiment to test whether the H567K Munc13-1 (Rhee Cell 2002), which is a phorbol ester binding deficient Munc13-1, abolishes the inhibitory effect of phorbol ester in chromaffin cells.

4. It appears that PMA itself can translocate Muncs to the plasma membrane (Figures 4 &7), which might be more potent than calcium-mediated membrane targeting Figure 6 vs. 7. Several questions: The expression level of protein expression mediated by the Smiliki viruses is very difficult to control. For data shown in figure 4A, are those fluorescence aggregates inside the cell? Moreover, can the author show time-lapse images of the translocation of Muncs to membrane-mediated by PMA? Please comment.

5. To track Munc13 translocation the authors have chosen EGFP-tagged variants which overlap in the emission with the standard FuraII/Furaptra emission. Consequently, the authors omitted Ca^2+^-imaging in these experiments and thereby lost crucial information regarding the development of [Ca]I before and after the uncaging flash. These parameters are of central importance for the Ca^2+^-dependent priming and exocytosis timing, respectively. This is particularly worrisome, because in several experiments with Munc13 expression hardly any RRP component is apparent in the displayed capacitance traces, which may indicate insufficient Ca^2+^-dependent vesicle priming (Figure 4). Under proper calcium control, both Ashery et al. 2000 (Figure 2) and Betz et al. 2001 (Figure 6) reported that Munc13-1 overexpression in wt chromaffin cells causes at least a 300% increase in the size of the EB compared to wt cells. Performing the same experiment, but without calcium imaging, the authors in Figure 4-Sup1 show hardly any increase in the size of the EB (violet trace Figure 4-Sup1) but a rather strong increase in the sustained phase of exocytosis, a phenotype that could be a result of low intracellular pre-flash calcium levels leading to insufficient vesicle priming. Unfortunately, the authors have not chosen other red-shifted protein tag to prevent such uncertainties. Furthermore, the display of the capacitance traces in several figures does not allow the appreciation of changes in the EB size or its components (e.g. RRP). This is a limitation of this study, but we realize that it would take considerable work to repeat the experiments with a different protein tag. At the minimum, the limitations of this study should be clearly stated.

6. As central hypothesis, the authors propose that they have identified a unique stimulatory triad of ubMunc13-2, Syt7 and DAG/phorbolesters, which is needed for dense core vesicle priming and fusion. For example, in contrast to the behavior of wt cells (e.g. Figure 1A) phorbolester treatment becomes inhibitory in cells lacking Syt7 and expressing ubMunc13-2 (Figure 7). Nonetheless, previously published data by Sorensen's group, obtained under similar preflash [Ca]I conditions (Tawfik et al., 2021; Figure 6—figure supplement 2 E-H), clearly show that PMA strongly potentiates exocytosis even in the absence of Syt7. Therefore, these previous findings by Tawfik et al. clearly counter the central hypothesis of the manuscript. Please clarify these disparate results.

Essential revisions related to Statistics and Analysis:

7. The choice of statistical analyses should be reconsidered. For example, they used non-parametric Mann-Whitney tests for most of the data but did not use the student t-test. In figure 2, they used the Kruskal-Wallis test, which is a one-way ANOVA but they have genotype differences and also the effect of PMAs, two independent variables. Please justify the use of these statistical tests. (Note: a useful reference on statistical analysis is Ho et al. "Moving beyond P values: data analysis with estimation graphics" Nature Methods 16, 565-566, 2019).

8. Related to the previous point, some of the data appear not normally distributed, but in the text and the figures, the mean was provided. Median and box plots would be more appropriate.

9. For imaging analysis, it is unclear how the membrane portion was determined. How do the authors determine the inside intensities? How to choose the confocal images to quantify the integrated density shown in figure 1M?

10. The authors speculate about the possibility that PMA treatment PMA-treatment of Unc13b KO cells may lead to spontaneous release, depleting the cells of secretory vesicles. To test this, they determined the integrated CgA-fluorescence over the entire cell (Figure 1M, N). By analyzing submembrane CgA-fluorescence, it should be possible to focus on a potential subcellular depletion of release-ready vesicles. Please perform this analysis.

11. After showing a detailed analysis of the exocytotic burst components and their kinetics in Figure 1 and 2 the authors argue on page 9 Line 275 'Since the measurements above indicated that the main effect of PMA is on secretion amplitude, not kinetics (see also (Nagy et al., 2006)), we only distinguished between burst secretion (first 1s secretion after Ca^2+^ uncaging, corresponding approximately to RRP and SRP fusion) and sustained secretion (last 4 s of secretion), as well as total secretion (the sum of burst and sustained release). '. However, the expression of Munc13 paralogs apparently leads to changes in the burst components and/or its kinetics (e.g. Figure 4B compare to Figure 1 or 2). In fact, these differences cannot be directly compared, because experiments like in Figure 3 and 4 lack the littermate wt control without and with PMA. Please comment.

12. Munc13 expression leads to a disproportionate increase in the sustained phase of release, which is not present with PMA. Please include detailed analyses of the exocytotic burst components and their kinetics to address these uncertainties.

---

## [Author Response]

Essential revisions:1. The conserved domain architectures of Munc13-1 and ubMun13-2 are remarkably similar. Thus, the results from this study are surprising. Please comment and speculate how the different, but closely related Munc13 paralogs can have such different mechanisms.

We have expanded a paragraph in the Discussion to answer this point.

2. It is claimed that " plasma membrane targeting of Munc13-1 is inhibitory for chromaffin cell DV secretion" (line 498). However, overexpression of Munc13-1 facilitates the release in wild-type cells (Figure 4-S1). Should one expect this overexpressed Munc13-1 not to increase the Ca^2+^-dependent membrane targeting?

We have attempted to clarify by changing the statement to “plasma membrane targeting of Munc13-1 by PMA is inhibitory.”. The context of the statement is the discussion of why Doc2B expression in Unc13b KO cells is inhibitory, which we suggest is because Doc2B will bring Munc13-1 to the plasma membrane (like PMA). Later in the Discussion, we discuss that Munc13-1 might exert its positive effect (in the absence of PMA) at the vesicular level, together with CAPS^-1^/2. Also note that we did not see much Ca^2+^dependent targeting of Munc13-1 in this investigation – at least not any that correlate with secretion.

3. The authors noted that in the Rosenmund et al. 2002 paper, they had shown that phorbol ester potentiated glutamate release (by more than twofold). Ideally, please perform an experiment to test whether the H567K Munc13-1 (Rhee Cell 2002), which is a phorbol ester binding deficient Munc13-1, abolishes the inhibitory effect of phorbol ester in chromaffin cells.

We have performed the experiments suggested by the reviewer, and present them in the new Figure 4 – supplementary figure 3. As expected, there is no inhibitory effect of phorbol esters in chromaffin cells expressing the H567K Munc13-1. This confirms previous data (Bauer et al., 2007) obtained in bovine chromaffin cells, which is important. We thank the reviewer for the suggestion.

4. It appears that PMA itself can translocate Muncs to the plasma membrane (Figures 4 &7), which might be more potent than calcium-mediated membrane targeting Figure 6 vs. 7. Several questions: The expression level of protein expression mediated by the Smiliki viruses is very difficult to control. For data shown in figure 4A, are those fluorescence aggregates inside the cell? Moreover, can the author show time-lapse images of the translocation of Muncs to membrane-mediated by PMA? Please comment.

The fluorescence aggregates are found inside the cell, and are present in some cells, but not all. We do not know the reason for this, but aggregates are also visible upon expression in HEK-cells (Betz et al., 1998 and own data now shown). We have followed the suggestion by the reviewer and included time-lapse images of a few cells expressing Munc13-1, Munc13-2 and Munc13-1 H567K in the new Figure 3—figure supplement 4. The number of experiments is not large, because we can only apply PMA once to a coverslip of chromaffin cells, as PMA cannot be washed out. Nevertheless, the experiment demonstrates the recruitment of Munc131 and ubMunc13-2, but not Munc13-1 H567K to the plasma membrane, which is a nice control. Note that the actual timing of recruitment (which we imaged for 800 s) is likely dominated by the time it takes for PMA to enter the cell, and thus it is unlikely to reflect a biologically important parameter.

5. To track Munc13 translocation the authors have chosen EGFP-tagged variants which overlap in the emission with the standard FuraII/Furaptra emission. Consequently, the authors omitted Ca^2+^-imaging in these experiments and thereby lost crucial information regarding the development of [Ca]I before and after the uncaging flash. These parameters are of central importance for the Ca^2+^-dependent priming and exocytosis timing, respectively. This is particularly worrisome, because in several experiments with Munc13 expression hardly any RRP component is apparent in the displayed capacitance traces, which may indicate insufficient Ca^2+^-dependent vesicle priming (Figure 4). Under proper calcium control, both Ashery et al. 2000 (Figure 2) and Betz et al. 2001 (Figure 6) reported that Munc13-1 overexpression in wt chromaffin cells causes at least a 300% increase in the size of the EB compared to wt cells. Performing the same experiment, but without calcium imaging, the authors in Figure 4-Sup1 show hardly any increase in the size of the EB (violet trace Figure 4-Sup1) but a rather strong increase in the sustained phase of exocytosis, a phenotype that could be a result of low intracellular pre-flash calcium levels leading to insufficient vesicle priming. Unfortunately, the authors have not chosen other red-shifted protein tag to prevent such uncertainties. Furthermore, the display of the capacitance traces in several figures does not allow the appreciation of changes in the EB size or its components (e.g. RRP). This is a limitation of this study, but we realize that it would take considerable work to repeat the experiments with a different protein tag. At the minimum, the limitations of this study should be clearly stated.

We have inserted a brief discussion of the difference between our data and those of Ashery and Betz in the beginning of the Discussion. It is very hard to compare our data obtained in newborn or embryonic mouse chromaffin cells directly to experiments performed a long time ago in adult bovine chromaffin cells (Ashery et al., 2000, Betz et al., 2001). None of those two papers reported (or – based on their use of low affinity fura-dyes – could have measured) the calcium-concentration in the cell before stimulation. Calcium dependent priming has been studied in detail primarily in mouse chromaffin cells (starting with Voets, 2000, Neuron 28: 537-545), and whether it works the same way in adult bovine cells is unclear. Therefore, there is a species/age difference, which makes it hard to compare to the previous work. It is also not clear whether the stimulation paradigm is really similar, since previous experiments sometimes made use of ‘prestimulation’ before the actual stimulus, to stimulate priming. We display the result of the first stimulation that the cell experienced.

With our method, we did not measure the prestimulation calcium concentration in each cell directly, but we established the range that we are working within in separate experiments. It is possible that calcium concentrations were a little lower than we thought, mainly because of uncertainties in the calcium calibration, which is an intrinsic limitation of calcium-experiments. However, both groups were measured in parallel experiments, on the same days. We clearly state that because of this design, some experiments might fall outside the estimated range. Because the findings in our manuscript were somewhat unexpected, we chose to use the exact same constructs that had been used previously, re-sequencing them to establish their identity. There is always a trade-off involved in these decisions, and we agree with the reviewer that using a red-shifted tag would have been elegant. We think that the limitations of our study do not put our conclusions in doubt.

6. As central hypothesis, the authors propose that they have identified a unique stimulatory triad of ubMunc13-2, Syt7 and DAG/phorbolesters, which is needed for dense core vesicle priming and fusion. For example, in contrast to the behavior of wt cells (e.g. Figure 1A) phorbolester treatment becomes inhibitory in cells lacking Syt7 and expressing ubMunc13-2 (Figure 7). Nonetheless, previously published data by Sorensen's group, obtained under similar preflash [Ca]I conditions (Tawfik et al., 2021; Figure 6—figure supplement 2 E-H), clearly show that PMA strongly potentiates exocytosis even in the absence of Syt7. Therefore, these previous findings by Tawfik et al. clearly counter the central hypothesis of the manuscript. Please clarify these disparate results.

Thank you for pointing out that the link to our previous paper was not clearly established in our manuscript. First, it was an overstatement to use the words ‘obligatorily’ and ‘unique’ for the functional pairing between phorbolester x ubMunc13-2 and syt7, since our previous paper showed – as the reviewer correctly points out – that in the absence of syt7 other mechanisms can kick in at higher prestimulation calcium concentration. We have softened the claims and we have added explicit discussion of the previous data in the Discussion section. Regarding the reduction in secretion upon application of PMA in Syt7 KO cells overexpression ubMunc13-2, the reduction in secretion is directly related to the increase in secretion seen when ubMunc13-2 is overexpressed in the Syt-7 KO. This increase is basically eliminated when PMA is added. Thus, ubMunc13-2 per se can increase secretion when overexpressed in the Syt-7 KO, but not in the presence of PMA. We have added discussion of this as well, comparing to the previous paper. We hope that this has clarified the situation.

Essential revisions related to Statistics and Analysis:7. The choice of statistical analyses should be reconsidered. For example, they used non-parametric Mann-Whitney tests for most of the data but did not use the student t-test. In figure 2, they used the Kruskal-Wallis test, which is a one-way ANOVA but they have genotype differences and also the effect of PMAs, two independent variables. Please justify the use of these statistical tests. (Note: a useful reference on statistical analysis is Ho et al. "Moving beyond P values: data analysis with estimation graphics" Nature Methods 16, 565-566, 2019).

We thank the reviewer for the useful reference and note that by displaying all data points, we follow a major recommendation of that paper. We have not inserted the δ-metric, because we feel this would overly burden already complex figures. Besides, it seems most meaningful for normally distributed data, and some of our data are not normally distributed. Regarding the choice of test, most of our tests are two-sample, and, indeed, there is always a question about whether to use t-test or Mann-Whitney test in this case. In the past, we (and others) have often resorted to a complex decision sequence, where we would first test whether the requirements for doing a t-test are fulfilled, otherwise we would do log-transform, and finally (if nothing else worked), we would perform Mann-Whitney test. However, this is not very satisfactory, because it means that the hypothesis tested will vary between data sets. For instance, the Burst size might be tested with a ttest in one instance, and a Mann-Whitney test in another instance. This tests different hypotheses (differences in means, or median), which is almost never discussed, but could have non-intuitive consequences. Therefore, in an attempt to simplify things we decided to use non-parametric tests (MannWhitney or Kruskal-Wallis) in all cases, which relies on a minimum set of assumptions. However, unfortunately, the complicated decision sequence was still present in our Statistics section. It has now been deleted and the correct rationale has been inserted.

It is a very good observation that the data in Figure 2 could be arranged as a two-way ANOVA, with genotype and PMA application as the two orthogonal factors. In that case, our hypothesis essentially states that there is an interaction between drug (PMA) and genotype. Performing this 2-way ANOVA separately on the RRP amplitude and the SRP amplitude showed that the interaction was indeed close to significant in both cases (P=0.0562 for RRP and P=0.0442 for SRP), indicating a likely interaction between the two factors. Note that this actually fits very well with the Kruskall-Wallis tests and post-tests already performed, which showed that the SRP for Munc13-1 KO PMA comparing to Munc13-1 WT PMA was significantly different (i.e. higher) (p=0.0437), whereas the for the RRP the difference was non-significant. The two ways of analyzing the data do not seem test exactly the same hypothesis, though, since stating that there is an interaction between two terms is weaker/different than stating that one genotype has a different response than the other in the presence of drug – and another point is that the ANOVA is parametric, but the Kruskal-Wallis is nonparametric. We have therefore opted to add the ANOVA to the analysis and text, while keeping the original analysis as well.

8. Related to the previous point, some of the data appear not normally distributed, but in the text and the figures, the mean was provided. Median and box plots would be more appropriate.

We appreciate the point, and we have changed the display of all graphs to be box-and-whiskers with individual data displayed, so the entire distribution is still visible. We have also updated the source data for each figure with the median value. We still cite the mean ± SEM in the text – and they are still in the source data – because these values are often used in the field. Overall, we now cite mean, SEM, median and display all data for each data set, so all relevant information is available for the reader.

9. For imaging analysis, it is unclear how the membrane portion was determined. How do the authors determine the inside intensities? How to choose the confocal images to quantify the integrated density shown in figure 1M?

This is the procedure as stated (now clarified) in the Materials and methods: “Integrated densities of ROIs manually drawn around the entire cell (Total intensity) or excluding the rim of the cell (Inside intensity) were background subtracted. The Total intensity divided by the Inside intensity was used as a measure for plasma membrane localization.” For confocal images, we obtained a stack of images and for quantification used the image plane, where the cell diameter was largest. This information has been added to the Materials and methods.

10. The authors speculate about the possibility that PMA treatment PMA-treatment of Unc13b KO cells may lead to spontaneous release, depleting the cells of secretory vesicles. To test this, they determined the integrated CgA-fluorescence over the entire cell (Figure 1M, N). By analyzing submembrane CgA-fluorescence, it should be possible to focus on a potential subcellular depletion of release-ready vesicles. Please perform this analysis.

This is a good idea, and we have performed the analysis – it is included in the new Figure 1O. There are no significant differences.

11. After showing a detailed analysis of the exocytotic burst components and their kinetics in Figure 1 and 2 the authors argue on page 9 Line 275 'Since the measurements above indicated that the main effect of PMA is on secretion amplitude, not kinetics (see also (Nagy et al., 2006)), we only distinguished between burst secretion (first 1s secretion after Ca^2+^ uncaging, corresponding approximately to RRP and SRP fusion) and sustained secretion (last 4 s of secretion), as well as total secretion (the sum of burst and sustained release). '. However, the expression of Munc13 paralogs apparently leads to changes in the burst components and/or its kinetics (e.g. Figure 4B compare to Figure 1 or 2). In fact, these differences cannot be directly compared, because experiments like in Figure 3 and 4 lack the littermate wt control without and with PMA. Please comment.

It is true that there are changes in burst components after expressing Munc13 paralogs, but this is expected if the burst size is changed, which would be expected to change both bust components (RRP and SRP). It would not be visible from our plots of average capacitance changes whether time constants for fusion are changed. We decided to simplify the analysis, because Figure 1 and 2, as well as previous data, indicated that the major change by PMA is on the amplitude of secretion, whereas kinetics is not much affected. Moreover, our results (increase or decrease in overall secretion amplitude) does not require kinetic analysis. We therefore felt that presenting the full kinetic analysis of every experiment detracts attention from the main finding. We simplified the analysis to focus attention on the main findings. None of our conclusions are based on fusion kinetics. We probably do not fully understand the final comment (‘In fact… without and with PMA’). Comparison between Munc13-1-EGFP and Munc13-1-EGFP + PMA in Figure 4 clearly show a decrease by PMA, whereas experiments in wt control cells (from the same mouse line, i.e. *Unc13a*) in Figure 2 show an increase by PMA application. Thus, the difference is clear. We only plot experiments in the same graph (and compare them statistically) when they have been carried out in parallel experiments (i.e. using the same cultures on the same measurement days), and doing experiments in four different groups on every recording day is very cumbersome.

12. Munc13 expression leads to a disproportionate increase in the sustained phase of release, which is not present with PMA. Please include detailed analyses of the exocytotic burst components and their kinetics to address these uncertainties.

We are not sure what the reviewer refers to. As an example, we have now provided the full kinetic analysis of the Munc13-1 expressed in wt CD1 cells in Figure 4 —figure supplement 1. As stated above, (answer to point 11), we think that providing the full kinetic analysis in each case detracts attention from the main findings of our manuscript.